# Evolutionary Feature Engineering for Structured Data

Ege Onur Taga [1]   Yilin Zhuang [1]   M. Emrullah Ildiz [1]   Petros Mol [2]
Abhimanyu Das [2]   Karthik Duraisamy [1]   Samet Oymak [1 2]

## Abstract

Large language models are increasingly used as open-ended search operators in evolutionary optimization. We introduce Evolutionary Feature Engineering (EFE), a framework that uses LLM-based evolution to discover preprocessing programs for structured data. EFE represents transformations as Python programs with a standardized `fit`/`transform` interface and refines them using dataset context, summary statistics, and validation feedback. For time-series forecasting, EFE-Time learns invertible dataset-specific normalizations that improve time-series foundation models, reducing MASE, WQL, and MAE by at least 3% on average and up to 19% on COVID-Deaths. For tabular prediction, EFE-Tab evolves compact, interpretable feature programs that improve or match prior LLM-based feature-engineering methods, with especially strong gains for decision trees. Overall, EFE shows that LLM-based evolution can improve structured-data model performances.

## 1. Introduction

Feature engineering has a simple premise: a model need not change if the data can be presented in a better form. In forecasting, normalization or detrending can make a sequence easier to extrapolate; in tabular prediction, a ratio, interaction, or threshold can make a simple classifier substantially more expressive. Such transformations are often obvious in hindsight to experts but difficult to specify in advance, since they depend on the dataset, the downstream model, and structure not captured by fixed preprocessing libraries.

Large language models make it possible to search over a richer open-ended space of executable feature-engineering programs. Recent systems such as AlphaEvolve (Novikov et al., 2025) show that LLMs can guide evolutionary code

[1]University of Michigan [2]Google Research. Correspondence to: Ege Onur Taga <egetaga@umich.edu>.

*Proceedings of the 2nd ICML Workshop on Foundation Models for Structured Data*, Seoul, South Korea. 2026. Copyright 2026 by the author(s).

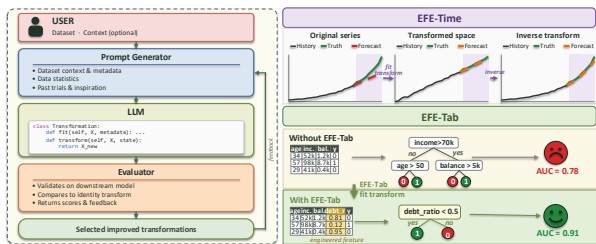

*Figure 1.* Given a dataset and optional context, EFE uses metadata, statistics, and past feedback to prompt an LLM to propose `fit`/`transform` programs. Candidates are evaluated against the identity baseline, and scores are fed back into an evolutionary loop.

discovery, while LLM-based feature-engineering methods such as CAAFE and LLM-FE show that dataset context can help construct useful tabular features (Hollmann et al., 2023; Abhyankar et al., 2025). These results suggest a broader question: can LLM-based evolution discover useful preprocessing programs for structured data, specifically time series and tables, where temporal order and column structure constrain and guide LLM's action space?

We propose *Evolutionary Feature Engineering* (EFE) as a framework for evolving feature transformations as Python programs. Each candidate follows a standard `fit`/`transform` interface and is evaluated by inserting it before a fixed downstream model. An LLM proposes transformations using dataset metadata, summary statistics, and outcomes of previous trials; validation performance is then returned to the evolutionary loop as feedback. Thus, search is driven directly by downstream task performance and dataset-specific context which captures domain expertise.

We instantiate EFE in two settings, as shown in Figure 1. For time-series forecasting, *EFE-Time* evolves dataset-specific, invertible normalization programs with `fit`, `transform`, and `inverse_transform` methods. This is motivated by the importance of normalization for time-series foundation models: fixed transforms such as RevIN (Kim et al., 2022) or arcsinh-style scaling (Ansari et al., 2025) are useful, but cannot handle all trends, scales, outliers, and seasonal patterns. EFE-Time instead tailors the transformation to each dataset. The forecaster operates in the transformed space, and predictions are mapped back before evaluation.

For tabular prediction, *EFE-Tab* searches for compact, high-

value feature programs. Rather than generating large feature sets, it optimizes validation AUC improvement over the raw-feature baseline while penalizing added or dropped features. This encourages transformations that justify their complexity, especially when the downstream model is intended to remain simple and interpretable.

Experiments show that EFE improves performance in both domains. On the synthetic exponential-trend forecasting task, EFE-Time discovers transformations that make series easier for foundation models to extrapolate. On real time-series datasets from GIFT-Eval (Aksu et al., 2024), EFE-Time improves Chronos-2 (Ansari et al., 2025) on several datasets and transfers to other forecasters, including TimesFM 2.5 (Das et al., 2024), Moirai 2.0 (Liu et al., 2026a), and Reverso (Fu et al., 2026). It is also complementary to model adaptation: applying the evolved transform before fine-tuning yields larger average gains than either component alone (Figure 4). On tabular datasets, EFE-Tab achieves the best mean rank among compared feature-engineering methods across TabPFN, LightGBM, and decision trees, with particularly strong gains for decision trees (see Table 2, Figure 5).

Overall, our contributions are threefold. **First**, we formulate feature engineering for structured data as evolutionary search over stateful preprocessing programs. **Second**, we introduce EFE-Time, which evolves invertible dataset-specific transformations for time-series foundation models. **Third**, we introduce EFE-Tab, a parsimonious method that balances predictive improvement against feature complexity. Due to space limits, we focus on EFE-Time in the main body and defer EFE-Tab and related work to Appendices B and D.

## 2. Evolving Feature Programs

EFE searches over executable preprocessing programs with a standard `fit`/`transform` interface, allowing the selected transformation to be inserted directly before an existing forecasting or prediction pipeline. EFE keeps the downstream model and training protocol fixed, changing only the input representation. Starting from the identity transformation, EFE runs an evolutionary loop. At each iteration, a prompt generator gives an LLM dataset context, metadata, summary statistics, prior feedback, and examples of strong programs. The LLM proposes a modified program, which the evaluator checks for executability, inserts before the fixed downstream model, and scores against the identity baseline. The resulting score and feedback guide later proposals. Thus, the LLM acts as a variation operator, while selection is driven by downstream validation performance.

### 2.1. General Setup

Let $\mathcal{D}$ be a structured dataset with optional context $c$. A validation protocol produces $K$ evaluation instances $\Pi(\mathcal{D}) =$

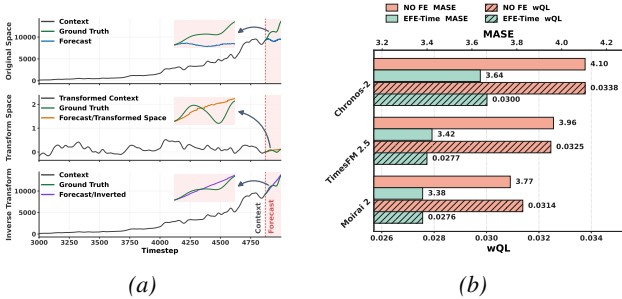

*(a)*                    *(b)*

*Figure 2.* (a) Chronos-2 under-extrapolates steep exponential growth; transforming the series improves forecast alignment, and inverse-transforming this yields a closer fit. (b) Synthetic exponential benchmark results: raw inputs (NO FE) vs. transformed forecasting with inversion (EFE-Time).

$\{(D_j^{\text{fit}}, D_j^{\text{in}}, Y_j)\}_{j=1}^K$, where $D_j^{\text{fit}}$ is the data available to a pre-processing program when fitting its state, $D_j^{\text{in}}$ is the input passed to the downstream routine, and $Y_j$ is the held-out target used only for evaluation. A candidate program $p$ consists of three operations: $\text{fit}_p$, $\text{transform}_p$, and $\text{post}_p$. On validation instance $j$, it is applied as $\sigma_{j,p} = \text{fit}_p(D_j^{\text{fit}}; c)$, $\quad \widetilde{D}_{j,p} = \text{transform}_p(D_j^{\text{in}}; \sigma_{j,p})$ $\quad \widehat{Y}_{j,p} = \text{post}_p\big(\mathcal{B}_j(\widetilde{D}_{j,p}); \sigma_{j,p}\big)$, where $\mathcal{B}_j$ is a fixed downstream predictive routine. EFE does not modify $\mathcal{B}_j$; it searches only over preprocessing programs $p$. For a lower-is-better loss $\ell$, we define $\widehat{L}(p) = \frac{1}{K} \sum_{j=1}^K \ell(Y_j, \widehat{Y}_{j,p})$. Let $p_{\text{id}}$ denote the identity preprocessing program. For loss-based tasks, we measure relative improvement as $\Delta(p) = 1 - \widehat{L}(p)/\widehat{L}(p_{\text{id}})$. For utility-based metrics such as AUC, we instead use additive improvement over $p_{\text{id}}$. The final score $s(p)$ combines validation improvement with reliability, runtime, and complexity penalties. Exact final scores for EFE-Time and EFE-Tab are provided in Sections 2.3 and B.1 The valid program class $\mathcal{P}_{\text{valid}}$ enforces executability, preservation of the required input structure, and leakage prevention: $\text{fit}_p$ may use only $D_j^{\text{fit}}$ and context $c$, while $Y_j$ is accessible only to the evaluator.

### 2.2. LLM-Driven Evolutionary Optimization

Following Liu et al. (2026b), we formalize the evolutionary optimization where EFE performs sequential search over $\mathcal{P}_{\text{valid}}$. After $t$ evaluations, the optimizer maintains the history $\mathcal{H}_t = \{(p_i, s_i, a_i)\}_{i=1}^t$, where $s_i = S(p_i)$ and $a_i$ contains auxiliary feedback such as logs, diagnostics, or evaluator feedback. At step $t$, a search strategy $S_t$ selects parent candidates, a prompt-level variation operator, and optional inspiration examples: $(pc_t, \pi_t, I_t) \sim C_{S_t}(\mathcal{H}_t)$. An LLM-based generator proposes a new candidate, $p_{t+1} \sim G_{\text{sol}}(\cdot \mid pc_t, \pi_t, I_t)$, which is evaluated and appended to the history $(s_{t+1}, a_{t+1}) = E(p_{t+1})$, $\quad \mathcal{H}_{t+1} = \mathcal{H}_t \cup \{(p_{t+1}, s_{t+1}, a_{t+1})\}$. Under budget $T$, EFE returns the best program in the final population $p^\star \in \arg\max_{(p,s,a) \in \mathcal{H}_T} s$. Thus, the search strategy determines which candidates are reused, how they are varied, and which prior examples are shown to the generator.

Table 1. Comparison of EFE-Time against identity transformation across time-series datasets for Chronos-2.

| Dataset | Baseline | | | EFE-Time | | | % Improvement | | |
|---|---|---|---|---|---|---|---|---|---|
| | MASE | WQL | MAE | MASE | WQL | MAE | MASE | WQL | MAE |
| CovidDeaths | 35.444 | 0.039 | 124.241 | **31.853 ± 0.835** | **0.032 ± 0.003** | **102.830 ± 13.968** | 10.1 ± 2.3 | 19.5 ± 8.7 | 17.2 ± 11.2 |
| SolarHourly | 0.996 | 0.352 | 12.934 | **0.915 ± 0.049** | **0.335 ± 0.002** | **11.883 ± 0.648** | 8.1 ± 4.9 | 4.7 ± 0.7 | 8.1 ± 5.0 |
| M4Yearly | 3.370 | 0.117 | 905.276 | **3.183 ± 0.135** | **0.115 ± 0.002** | 868.517 ± 26.300 | 5.6 ± 3.9 | 2.1 ± 1.6 | 4.1 ± 2.9 |
| JenaWeather (H) | 0.547 | 0.042 | 8.745 | **0.538 ± 0.004** | **0.041 ± 0.001** | **8.436 ± 0.199** | 1.6 ± 0.6 | 2.1 ± 1.3 | 3.5 ± 2.3 |
| SZ_TAXI_15T | 0.543 | 0.200 | 2.722 | 0.543 ± 0.000 | 0.200 ± 0.000 | 2.722 ± 0.001 | 0.0 ± 0.0 | 0.0 ± 0.0 | 0.0 ± 0.0 |
| bitbrains_storage/H | 0.993 | 0.815 | 384.949 | 0.993 ± 0.000 | 0.815 ± 0.000 | 384.949 ± 0.000 | 0.0 ± 0.0 | 0.0 ± 0.0 | 0.0 ± 0.0 |
| bitbrains_rnd/H | 5.821 | 1.032 | 175.961 | **5.818 ± 0.002** | 1.016 ± 0.022 | 168.439 ± 10.611 | 0.1 ± 0.0 | 4.3 ± 6.0 | 1.5 ± 2.1 |
| us_births_D | 0.345 | 0.017 | 234.437 | **0.342 ± 0.002** | **0.017 ± 0.000** | 232.163 ± 1.191 | 1.0 ± 0.5 | 1.0 ± 0.5 | 0.4 ± 0.2 |
| kdd_cup_2018_H | 0.999 | 0.396 | 24.712 | **0.971 ± 0.004** | **0.388 ± 0.002** | 24.281 ± 0.213 | 2.8 ± 0.4 | 2.0 ± 0.6 | 1.7 ± 0.9 |
| restaurant | 0.684 | 0.256 | 7.055 | **0.682 ± 0.002** | 0.256 ± 0.001 | **7.041 ± 0.016** | 0.3 ± 0.2 | −0.2 ± 0.2 | 0.2 ± 0.2 |
| **Mean % Improvement** | – | | | – | | | **3.0** | **3.6** | **3.7** |

## 2.3. Specialization to EFE-Time

For time-series forecasting, $\mathcal{D} = \{y_{1:T_i}^{(i)}\}_{i=1}^{M}$ and each validation instance is a rolling forecast window $j = (i, t)$ with history $y_{1:t}^{(i)}$ and future target $y_{t+1:t+H}^{(i)}$: $D_j^{\text{fit}} = y_{1:t}^{(i)}$, $D_j^{\text{in}} = (y_{1:t}^{(i)}, H)$, $Y_j = y_{t+1:t+H}^{(i)}$. The fixed downstream routine is a forecaster $F_\theta$. A candidate program transforms the history, the forecaster predicts in the transformed space, and the program maps the forecast back: $z_{1:t}^{(i)} = \text{transform}_p(y_{1:t}^{(i)}; \sigma_{j,p})$, $\widehat{z}_{t+1:t+H}^{(i)} = F_\theta(z_{1:t}^{(i)}, H, c^{(i)})\widehat{y}_{t+1:t+H}^{(i)} = \text{inverse\_transform}_p(\widehat{z}_{t+1:t+H}^{(i)}; \sigma_{j,p})$, Thus, $\text{post}_p \equiv \text{inverse\_transform}_p$. Because predictions are made in the transformed space, EFE-Time restricts search to approximately invertible programs. For example, a candidate may fit a recent median $m$ and robust scale $q$, transform $y$ to $(y - m)/q$, let the forecaster predict in that normalized space, and invert predictions as $qz + m$. Using MASE as the selection loss, $s_{\text{time}}(p) = \left(1 - \frac{\text{MASE}(p)}{\text{MASE}(p_{\text{id}})} - \lambda_\tau^{\text{time}} \tau(p)\right) \mathbf{1}\{p \in \mathcal{P}_{\text{inv}}\}$. Typical EFE-Time normalizations include robust scaling, variance-stabilizations, history-only detrending, seasonal adjustment, and changepoint-aware rescaling.

## 3. Experiments

We evaluate the selected programs $p^\star$ in two settings: time-series forecasting and tabular prediction. EFE-Time evolves approximately invertible programs $p$ for time-series foundation models, while EFE-Tab evolves compact feature programs for tabular classifiers. We focus on EFE-Time here, while EFE-Tab disucssion is in the Appendix B. Code is available at https://github.com/egetaga/EFE.

### 3.1. Experimental Setup

**Evolutionary Optimization Interface.** Both EFE-Time and EFE-Tab are implemented on top of OpenEvolve (Sharma, 2025), an open-source evolutionary program-search system inspired by AlphaEvolve (Novikov et al., 2025). We use its code-diversity coordinate as the

MAP-Elites feature and use Claude-Opus-4.6 as the LLM backbone in all main experiments. All runs start from $p_{\text{id}}$. At each iteration, the LLM receives parent code, selected inspiration programs, score metadata, dataset context, and aggregate evaluator feedback. The proposed program $p_{t+1}$ is executed, checked for validity, inserted before a fixed downstream model, and scored against $p_{\text{id}}$.

**EFE-Time implementation details.** For EFE-Time, each dataset has a fixed dataset context computed from the training split, summarizing metadata such as frequency, prediction length, seasonal period, number of evaluation series, and aggregate series statistics. Each iteration, valid candidates $p$ receive an evaluation_summary containing aggregate validation metrics for the identity baseline and candidate, metric ratios, error counts, timing information, and the fraction of evaluated series helped or harmed. Failed candidates receive the transformation errors. The prompt never includes raw time-series values, forecasts, per-series errors, or sampled evaluation subsets. Each candidate $p$ is a TransformProgram with fit, transform, and inverse_transform. At execution time, fit receives only one historical series $D_j^{\text{fit}}$ and summary metadata, returning $\sigma_{j,p}$ for the forward and inverse transformations.

**Benchmarks and evaluation.** We evaluate EFE-Time on ten GIFT-Eval datasets (Aksu et al., 2024) spanning healthcare, energy, finance, weather, transportation, and web/cloud operations. Programs are evolved against Chronos-2 (Ansari et al., 2025) using validation MASE improvement over $p_{\text{id}}$ as the selection signal. For each dataset and seed, we freeze the best program $p^\star$ and evaluate it on the held-out test split. To test cross-model transfer, we apply the same frozen $p^\star$, without re-evolution, to TimesFM 2.5, Moirai 2-Small, and Reverso-Nano (Das et al., 2024; Liu et al., 2026a; Fu et al., 2026). We run three seeds for 100 iterations and report mean MASE, wQL, and MAE with standard deviations.

### 3.2. EFE-Time Experiments

We first show that time-series foundation models struggle on synthetic exponential-trend data and that EFE-Time miti-

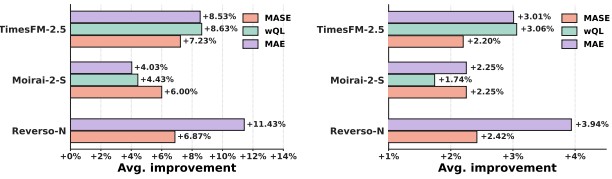

*(a)* Over three datasets    *(b)* Over entire benchmark set

*Figure 3.* Model transfer of EFE-Time: programs evolved on Chronos-2 and applied unchanged to TimesFM-2.5, Moirai-2-Small, and Reverso-Nano.(a) Mean improvement on Covid Deaths, M4-yearly, and solar hourly.(b) Across 10 GIFT-Eval datasets.

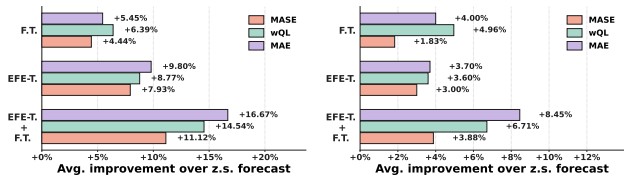

*(a)* Over three datasets    *(b)* Over entire benchmark set

*Figure 4.* Gains over Chronos-2 zero-shot forecasts. (a) Mean percent improvement on Covid Deaths, M4-yearly, and solar hourly. (b) Mean across 10 GIFT-Eval datasets. EFE-Time plus fine-tuning (EFE-T.+F.T.) consistently yields the largest gains.

gates this failure mode.

### 3.2.1. TSFMs Struggle with Multiplicative Growth

We generate synthetic time series with smooth exponential growth and no spikes or abrupt events, following Appendix F.6. Each sequence is built from a random initial value and growth factor, yielding a monotone latent trend with smooth multiplicative noise. We sample 100 sequences of length 5,000 and reserve the final 128 steps for testing. Chronos-2, TimesFM-2.5, and Moirai-2.0 perform poorly on this multiplicative-growth data despite its smoothness as shown in Figure 2b. But, EFE-Time program $p^\star$, optimized using Chronos-2, substantially improves performance across all TSFMs. Figure 2a shows why: $p^\star$ maps the series to an approximately stationary space where Chronos-2 forecasts more accurately, then inverts the transformation to recover a forecast that closely matches the ground truth.

### 3.2.2. Dataset Specific Normalizations Improve TSFMs

**Learning Normalization Programs for Chronos 2.** We evaluate EFE-Time on real datasets from GIFT-Eval (Aksu et al., 2024), using Chronos-2 as the downstream forecaster. As shown in Table 1, EFE-Time improves over raw Chronos-2 forecasts, i.e., $p_{id}$, by 3.0%, 3.6%, and 3.7% on average in MASE, WQL, and MAE. Gains vary across datasets: Covid Deaths, Solar (hourly), and M4-Yearly improve by up to 19%, while `kdd_cup_2018` improves by about 3% and several datasets show modest or statistically insignificant gains. This is expected, since normalization is not equally helpful when datasets are regular or remain difficult after transformation. We visualize representative raw, transformed-space, and inverted forecasts in Figures 10, 11, 12, 13, and 14.

**Model Transfer of Learned Normalization Programs.** We investigate whether the learned dataset-specific normalizations transfer to other time-series foundation models. To this end, we evaluate TimesFM-2.5, Moirai-2-small (the other Moirai-2 variants are not open-sourced), and Reverso-Nano (Fu et al., 2026). We chose Reverso-Nano because it is extremely lightweight compared to the other TSFMs considered, allowing us to investigate whether the improve-

ments persist for smaller TSFMs as well. We use the same programs learned using Chronos-2, without modification.

As shown in Figure 3, the benefits of learned normalization programs transfer to other TSFMs. Specifically, on Covid Deaths, M4-Yearly, and Solar (hourly), shown in Figure 3a, we observe an average MASE improvement of more than 6% across all TSFMs, reaching 7% for TimesFM. Over the entire benchmark set, shown in Figure 3b, we again observe consistent MASE improvements of 2.20%, 2.25%, and 2.42% for TimesFM, Moirai-2-small, and Reverso-Nano, respectively. Overall, the consistent improvements achieved by these TSFMs on the same datasets, using programs learned with Chronos-2, suggest that the forecasting difficulty of these datasets is not specific to a particular model architecture. Rather, it arises from the underlying time-series dynamics, which EFE-Time makes more amenable to forecasting.

**Fine-tuning and Normalizations are Additive.** Both fine-tuning and learning EFE-Time programs use the training portion of the data. In Figure 4, we investigate how they compare and whether their benefits are additive; that is, whether first learning normalizations and then fine-tuning the model in the normalized space yields substantial gains. On datasets where normalization is highly effective, Figure 4a demonstrates that EFE-Time outperforms fine-tuning on every metric. On the entire benchmark set, shown in Figure 4b, we observe that EFE-Time provides larger gains than fine-tuning for MASE, but slightly lower gains for wQL and MAE. Importantly, as shown in Figure 4, EFE-Time+fine-tuning yields additive benefits in both settings, outperforming either approach.

### 3.3. Discussion

We introduced EFE, an evolutionary framework for refining feature-engineering programs. EFE-Time improves TSFM forecasting and is complementary to fine-tuning, while EFE-Tab learns parsimonious feature programs. Future work should study how evolutionary hyperparameters and feedback design affect performance, since both likely shape the quality of discovered programs.

## Acknowledgments

We thank Halil Alperen Gozeten of the University of Michigan for discussions on LLM-based evolutionary discovery. This work was supported by the National Science Foundation under grants CCF-2046816, CCF-2403075, and CCF-2212426; by the Office of Naval Research under grant N000142412289; and by Los Alamos National Laboratory under grant AWD026741 at the University of Michigan. The computational aspects of this research were generously supported by resources provided through the Amazon Research Award on Foundation Model Development.

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

# A. Discussion

We introduced EFE, an evolutionary framework for feature engineering that iteratively refines feature programs. EFE-Time improves the forecasting performance of TSFMs across diverse downstream applications and achieves performance competitive with fine-tuning. More importantly, combining EFE-Time with fine-tuning yields additive gains, suggesting that learned normalizations and model adaptation provide complementary benefits. EFE-Tab, in turn, learns compact and parsimonious feature-engineering programs that substantially improve simple decision trees while preserving interpretability. Our paper also points to several promising directions for future work. First, the evolutionary process currently uses fixed hyperparameters, such as the exploration–exploitation ratio or the number of evolutionary islands. This design keeps the experimental protocol simple and controlled, but future work could study how these choices affect performance. Second, the quality and structure of feedback play an important role in guiding program evolution. A detailed ablation of which feedback components are most useful, and when they help or hinder refinement, would provide a clearer understanding of the mechanisms driving EFE's improvements.

# B. EFE-Tab

## B.1. Specialization to EFE-Tab

For tabular prediction, $\mathcal{D} = \{(x_i, y_i)\}_{i=1}^{n}$, and each validation instance is a fold $(I_j^{\text{tr}}, I_j^{\text{val}})$:

$$D_j^{\text{fit}} = (X_{I_j^{\text{tr}}}, y_{I_j^{\text{tr}}}), \qquad D_j^{\text{in}} = (X_{I_j^{\text{tr}}}, y_{I_j^{\text{tr}}}, X_{I_j^{\text{val}}}), \qquad Y_j = y_{I_j^{\text{val}}}.$$

A candidate program fits its state on the training fold and applies the same state to both training and validation features:

$$\sigma_{j,p} = \text{fit}_p(X_{I_j^{\text{tr}}}, y_{I_j^{\text{tr}}}; c), \qquad \widetilde{X}_{I_j^{\text{tr}}, p} = \text{transform}_p(X_{I_j^{\text{tr}}}; \sigma_{j,p}), \qquad \widetilde{X}_{I_j^{\text{val}}, p} = \text{transform}_p(X_{I_j^{\text{val}}}; \sigma_{j,p}).$$

The fixed learner $\mathcal{A}$ is trained on the transformed training fold and evaluated on the transformed validation fold:

$$f_{j,p} = \mathcal{A}\left(\widetilde{X}_{I_j^{\text{tr}}, p}, y_{I_j^{\text{tr}}}\right), \qquad \widehat{y}_{I_j^{\text{val}}, p} = f_{j,p}(\widetilde{X}_{I_j^{\text{val}}, p}).$$

Here $\text{post}_p \equiv \text{id}$ because predictions are already in the original target space.

For binary classification, EFE-Tab uses mean AUC improvement over the identity program:

$$\bar{\delta}(p) = \frac{1}{K} \sum_{j=1}^{K} \left[ \text{AUC}_j(p) - \text{AUC}_j(p_{\text{id}}) \right].$$

The final score is

$$s_{\text{tab}}(p) = \bar{\delta}(p) - \lambda_f \sqrt{\frac{k(p)}{n}} - \lambda_\tau^{\text{tab}} \tau(p),$$

where $k(p)$ counts generated and dropped features, and $\tau(p)$ is runtime. This favors feature-engineering programs whose performance gains justify their added complexity.

## B.2. EFE-Tab implementation details.

For EFE-Tab, each iteration forms a validation instance $j$ by splitting the training data into a fitting pool (Pool A) and held-out scoring pool (Pool B). The dataset context is computed from Pool A and includes the dataset name, shape, target column, problem type, class balance, task description, column statistics, correlations, and a small number of sample rows. Because Pool A rotates across iterations, these summaries may vary slightly during evolution.

EFE-Tab evolves the `fit/transform` logic of a `FeatureProgram`. Programs learn $\sigma_{j,p}$ only from Pool A and apply the same fitted state to both Pool A and Pool B. Candidates may create interactions, ratios, nonlinear transforms, binned features, target encodings, group aggregates, or threshold indicators, and may also drop weak or redundant original columns.

Candidates are first checked for required entry points, valid outputs, schema consistency, and absence of in-place dataframe mutation. Valid programs are then scored on Pool B with $\mathcal{A}$ set to TabPFN: the model is fit on raw Pool A features to obtain the $p_{\text{id}}$ ROC-AUC on Pool B, and then refit on the candidate-engineered Pool A features to obtain the candidate ROC-AUC

*Table 2.* Comparison of feature engineering methods for single decision trees across tabular datasets over ROC-AUC. The reported results are 3 outer-fold averages from the TabArena benchmark.

| Dataset | Feature Engineering Method | | | |
|---|---|---|---|---|
| | **No FE** | **CAAFE** | **LLM-FE** | **EFE-Tab** |
| Churn | .8944 ± .0178 | .9239 ± .0181 | .9080 ± .0213 | **.9266 ± .0108** |
| HRAnalyticsJobChange | .7898 ± .0069 | .7864 ± .0093 | .7914 ± .0065 | **.7923 ± .0041** |
| E − CommerceShippingData | .7417 ± .0037 | .7441 ± .0034 | .7433 ± .0032 | **.7448 ± .0035** |
| in_vehicle_coupon_recommendation | .7296 ± .0140 | .7285 ± .0124 | .6720 ± .1042 | **.7473 ± .0134** |
| online_shoppers_intention | .9235 ± .0068 | .9242 ± .0041 | **.9382 ± .0133** | .9235 ± .0040 |
| Bank_Customer_Churn | .8340 ± .0035 | .8409 ± .0077 | **.8502 ± .0096** | .8434 ± .0040 |
| BankMarketing | .7329 ± .0056 | .7339 ± .0055 | .7345 ± .0048 | **.7348 ± .0064** |
| Diabetes | .7974 ± .0185 | .7852 ± .0473 | .7909 ± .0112 | **.8136 ± .0082** |
| Fitness_Club | .8042 ± .0118 | .7943 ± .0071 | .7971 ± .0105 | **.8048 ± .0128** |
| **Mean Rank** | 3.17 | 3.00 | 2.44 | **1.39** |

on transformed Pool B. The evaluator reports to the LLM the baseline and candidate AUCs, $s_{\text{tab}}(p)$, generated columns, dropped columns, output-column counts, and validation status. The evaluator also returns interpretable feedback to the LLM, computed only from Pool A: permutation-importance labels for surviving features, correlations between features in the table, a readable decision tree trained on the candidate representation, and an original-column importance report.

## B.3. EFE-Tab Benchmarks

We evaluate EFE-Tab on nine binary-classification datasets from TabArena (Erickson et al., 2026), spanning telecommunication churn, employment, e-commerce, banking, healthcare, and sports. During evolution, TabPFN-v2 (Grinsztajn et al., 2025) is the scoring model. After evolution, the selected programs are frozen and evaluated with TabPFN-v2 (Grinsztajn et al., 2025), LightGBM, and single decision trees. To control LLM API cost, we use the three folds from the first official repetition, run EFE-Tab for each fold for 100 iterations, and report averages and standard deviation over folds. We ran CAAFE and LLM-FE in the same way for 100 iterations for each fold.

As API costs prevented us from conducting experiments on the entirety of Gift-Eval and TabArena, we used selected datasets from these benchmarks, chosen to ensure domain diversity.

## B.4. EFE-Tab Experiments

**Learning Parsimonious Feature Programs.** Learning feature-engineering programs for tabular data with LLMs has been widely studied (Hollmann et al., 2023; Abhyankar et al., 2025; Nam et al., 2024). Thus, our main goal here is to show that the same EFE paradigm can be used straightforwardly to learn parsimonious feature-engineering programs, where each added or removed feature must yield a benefit greater than its associated penalty. We show that this approach improves decision trees while providing a significant interpretability advantage in real-world use.

We evaluate EFE-Tab learned feature engineering programs in Table 2 and compare it with no feature engineering, CAAFE, and LLM-FE, on a single decision tree. We see that EFE-Tab yields mean rank of 1.39 among 4 approaches when used with a single decision tree with the closest method after it achieving a mean rank of 2.44. We see in Figure 5 that EFE-Tab on our tabular benchmark also outperforms LLM-FE, and CAAFE on LightGBM and TabPFN downstream models. The seperation on LightGBM is high, whereas on TabPFN, each method achieves a similar mean rank. During experiments, we have observed that due to explicit penalty of parsimony, EFE-Tab yields compact feature programs, whereas LLM-FE and CAAFE usually generates higher number of feature interactions. We hypothesize that TabPFN has robustness for redundant features, yet a single decision tree, and also LightGBM has less so. Thus, we see a lower seperation between parsimonious feature engineering programs of EFE-Tab and competing approaches in TabPFN.

**The Regimes where EFE-Tab is Useful.** In Figure 6a, we vary the decision tree depth and observe that EFE-Tab generated programs have particularly strong benefit on shallower decision trees. Yet, the seperation exists in higher depths too, and more importantly only with depth 3, EFE-Tab achieves better performance than the no feature engineered baseline with higher depths. This signals that EFE-Tab is particularly good for learning simple but effective decision rules, which are particularly important for real-world applications requiring an explanation of the final decision.

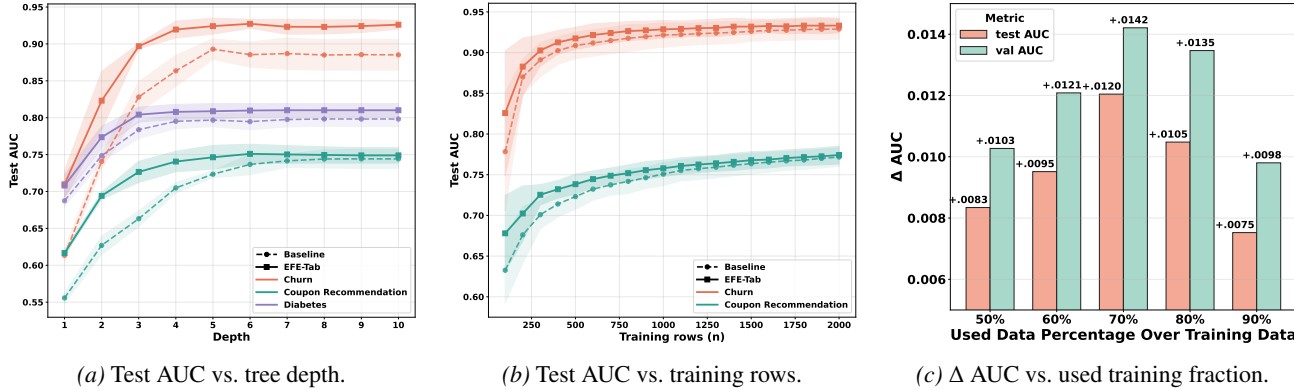

*(a)* Test AUC vs. tree depth.      *(b)* Test AUC vs. training rows.      *(c)* Δ AUC vs. used training fraction.

*Figure 6.* (a) Single decision tree over 3 folds × 5 seeds (15 runs per cell). (b) Test AUC vs. training-set size for TabPFN. EFE-Tab features deliver a larger lift in the low-data regime, with the gap shrinking as more training data becomes available. (c) Average Δ AUC (EFE-Tab - baseline) over the held-out test and validation sets vs. the fraction of training data exposed to the evolution loop, averaged over TabPFN/LightGBM/decision-tree and 3 seeds on *churn* and *in_vehicle_coupon_recommendation*.

In Figure 6b, we quantify the effect of EFE-Tab-generated programs in low-data regimes. We use TabPFN here because it performs well in low-data regimes and does not require hyperparameter tuning, which we observed to be unstable for decision trees and LightGBM when only small amounts of training data are available. We observe that, with less training data, feature engineering significantly boosts the performance of TabPFN. As the amount of data increases, this separation diminishes, demonstrating that generating meaningful features is particularly helpful for models in low-data scenarios.

**Characterizing the Fraction of Training Data Used.** As described in Section B.2, at each iteration, we randomly split the training data into pool A and pool B. Pool A summary is provided to the LLM and used as input to the downstream model, while pool B is used for scoring. This split naturally

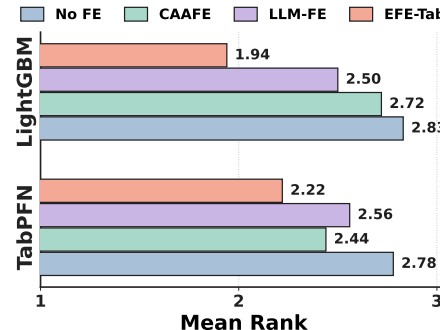

*Figure 5.* Mean rank of methods across our tabular benchmark.

introduces diversity across iterations. Figure 6c shows that EFE-Tab improvements follow an inverted-U trend as the fraction assigned to pool A increases. Large pool A fractions reduce diversity across iterations, whereas very small fractions make pool A less representative of the underlying dataset.

## C. EFE-Time

### C.1. Learning EFE-Time Programs Requires Strong LLMs.

Appendix F.8 evaluates EFE-Time with the Qwen3.5 family (Yang et al., 2025) on Covid-Deaths and Solar Hourly. Since these models are substantially weaker than Opus-4.6, we ran each experiment for 300 iterations. Qwen models perform reasonably well on Solar Hourly but are much less stable on Covid-Deaths. Given the difficulty of generating full transform and inverse-transform pipelines, this performance drop is expected. We believe that adapting EFE-Time to smaller, non-frontier LLMs remains a promising direction.

## D. Extended Related Work

**Classical Automated Feature Engineering Methods.** Early work on automated feature engineering for tabular data has largely relied on predefined transformation operators combined with search or selection strategies. Deep Feature Synthesis generates features from relational data by applying aggregation and transformation primitives (Kanter & Veeramachaneni, 2015), whereas Cognito uses a greedy search procedure to explore feature transformations that improve supervised learning performance (Khurana et al., 2016). AutoFeat constructs and selects nonlinear feature combinations (Horn et al., 2019), while learning-based methods, including LFE and reinforcement-learning approaches, aim to automatically identify useful transformations (Nargesian et al., 2017; Zhang et al., 2019). Despite their effectiveness, these methods typically make

limited use of rich dataset-level context, such as column semantics or task descriptions, and are instead primarily guided by fixed operators and heuristic search procedures.

**Evolutionary Optimization With LLMs.** Recent studies have investigated the use of LLMs to guide evolutionary optimization for discovering programs, prompts, and algorithms. FunSearch evolves candidate functions by combining LLM-generated proposals with evaluator feedback (Romera-Paredes et al., 2024), while AlphaEvolve extends this paradigm to full-program optimization through iterative code modification and evaluation (Novikov et al., 2025). OpenEvolve offers an open-source framework for this code search (Sharma, 2025). Other approaches further adapt the search process itself: EvoX evolves both candidate solutions and search strategies (Liu et al., 2026b), and AdaEvolve leverages LLM feedback for adaptive zeroth-order optimization (Cemri et al., 2026). LLM-based search has also been applied to scientific software generation (Aygün et al., 2025) and reflective prompt evolution (Agrawal et al., 2025). However, these works primarily focus on general-purpose program, prompt, or algorithm discovery, rather than feature engineering for structured data.

**Evolutionary Feature Engineering for Structured Data with LLMs.** More recent work has explored the use of LLMs for automated feature engineering on structured data. CAAFE iteratively prompts an LLM with dataset descriptions to propose semantically meaningful feature interactions and selects features one at a time based on changes in cross-validation performance (Hollmann et al., 2023). OCTree combines LLM reasoning with feedback from shallow decision trees fitted to the data, a direction that also informs our use of tree-based dataset reasoning (Nam et al., 2024). Most closely related to our work, LLM-FE adopts a FunSearch-style evolutionary search framework that incorporates dataset context to optimize tabular feature-transformation programs (Abhyankar et al., 2025; Romera-Paredes et al., 2024). However, LLM-FE evolves a single transformation function, rather than an AlphaEvolve-style full program that includes state fitted on the training data and an explicit inverse transformation (Novikov et al., 2025). In the time-series setting, ELATE uses evolutionary LLM search to generate predictive covariates, but it does not focus on invertible normalizations, and no open-source implementation is available for direct comparison (Murray et al., 2025). To the best of our knowledge, our work is the first to evolve invertible time-series normalization modules.

## E. Broader Impact

We develop a methodology for LLM-based evolutionary feature engineering. A potential risk of using LLMs for automated feature engineering is that they may introduce bias into the preprocessed data representations. This risk is relatively limited for EFE-Time, which learns invertible time-series normalizations, but is more pronounced for EFE-Tab, which operates directly on tabular features. In particular, semantic cues from column names may lead the LLM to generate spurious or discriminatory features. Therefore, EFE-Tab should be supervised carefully, and generated preprocessing programs should only be used after human review.

## F. Additional Experimental Details

We gives reproducibility details that are not explicit in the main text in Section 3.1. We also provide our code as part of the supplementary material.

### F.1. Evolutionary optimization interface

The OpenEvolve-specific configuration uses its island population and MAP-Elites archive, with code-diversity coordinates as the archive descriptor. Candidate proposals are complete program rewrites rather than parameter updates. The optimizer records both valid and invalid attempts so that later prompts can use failures as debugging feedback, but only valid programs are eligible for final selection. After the evolution budget is exhausted, the selected source code is frozen and reloaded by a separate held-out evaluator.

Prompts contain the parent program, a small set of previous programs or summaries selected by the optimizer, dataset-level context, and aggregate evaluator feedback. They do not contain evaluator source code, private scoring logic, held-out examples, or test-set results.

### F.2. EFE-Time Evolution

**Evolution Design.** Each time-series candidate implements `fit`, `transform`, and `inverse_transform`. At a forecast origin, `fit(y_hist, meta)` receives a single observed history and a compact metadata dictionary. The metadata keys are:

prediction length, seasonal period, history length, number of valid observations, mean, standard deviation, median, minimum, maximum, range, positive fraction, zero fraction, skewness, coefficient of variation, trend strength, recent mean, and recent standard deviation. The returned state dictionary must include `hist_len`. This is required because `inverse_transform` is called on forecast-horizon arrays rather than on the original history; any detrending, time-varying scaling, or seasonal-phase operation must therefore align forecast positions using the stored history length.

The validity gate checks the forward and inverse transforms before forecasting. A valid transform must preserve length, order, and NaN locations; avoid Inf values or new NaNs; remain numerically stable on forecast medians and quantiles; and avoid mapping nonconstant histories to constants. The inverse must be well defined on arrays of arbitrary forecast length, not only on arrays of length `hist_len`.

The prompt-visible `dataset_context` is fixed once per dataset from the training split. It contains only aggregate information such as frequency, horizon, seasonal period, number of evaluation series, and distributional summaries of histories. The per-iteration `evaluation_summary` is also aggregate: valid candidates receive baseline and candidate metrics, metric ratios, valid-series counts, error counts, timing, and help/harm rates; invalid candidates receive capped transformation or inverse-transformation errors. The LLM never sees raw series values, future values, sampled evaluation windows, model forecasts, per-series errors, baseline forecasts, scoring code, or test-set information.

**Hyperparameters.** All experiments use fixed Openevolve hyperparameters. We used Openevolve evolution with 3 islands, with exploitation ratio of 0.70, exploration ratio of 0.10, and elite selection ratio of 0.20. We give the system prompt we have used at Section J.1. We have a code diversity axis with 8 bins. We have used Claude Opus-4.6 through AWS API access with temperature of 0.7. While calculating the combined score, to let invalid programs live so that they could be fixed, we added an offset of 0.5 to the combined score so that bad performing programs won't get the same score as failing ones through validitiy checks. Therefore, an identity program corresponds to a combined score of 0.5. For each dataset, we extracted 3 validation portions from the training portion of the data with non-overlapping windows, using the given forecast length of the data. At each Openevolve iteration, we use 1/3 of total extracted samples, matching the original dataset size. For fine-tuning experiments with Chronos-2, we have used the official Chronos script from their github. We used Chronos in its univariate mode, forecasting each time series independently.

### F.3. EFE-Tab Evolution

**Evolution Design.** Beyond the Pool A/Pool B protocol described in Section 3.1, the tabular evaluator exposes a structured `dataset_context` computed only from Pool A. It includes the dataset name, shape, target column, problem type, class balance, task description, column-level statistics, target correlations, top inter-feature correlations, and a small number of Pool A rows. For numerical columns, the context reports ranges, quartiles, skewness, cardinality, and missingness, while for non-numerical columns, it reports cardinality, top values, and missingness. Since Pool A rotates across iterations, these summaries vary slightly during evolution.

The stage-one validity gate checks that the program has valid function signatures, returns DataFrames for both pools, preserves row alignment, avoids in-place mutation of the input frames, and produces a schema that can be applied consistently to Pool A and Pool B. Target-dependent operations such as target encodings or group aggregates are permitted only through state fitted on Pool A and then reused unchanged on Pool B.

For valid candidates, `stage2_summary` reports the baseline AUC, candidate AUC, AUC difference, combined fitness score, pool sizes, iteration seed, newly created columns, dropped original columns, output-column counts, and validation status. The evaluator may also return Pool-A-only interpretability feedback, including permutation-importance labels, an original-column importance report, and a readable decision tree trained on the candidate representation. Pool B is visible to the LLM only through aggregate AUC values and their difference. Raw or transformed Pool B rows, per-row predictions, per-row losses are not exposed.

The complexity term counts both added columns and dropped original columns as feature modifications. This makes the objective prefer feature programs whose held-out AUC gain is large enough to justify their additional representation cost and runtime.

**Hyperparameters.** Just like EFE-Time, all experiments use fixed Openevolve hyperparameters. We used Openevolve evolution with 3 islands, with exploitation ratio of 0.70, exploration ratio of 0.10, and elite selection ratio of 0.20. Thus, all the hyperparameters with respect to Openevolve are the same with EFE-Time. During evolution, we used 70% of the training data for Pool A, and the rest 30% of the data for pool B. At each Openevolve iteration, pool A and pool B are

resampled again.

## F.4. Benchmarks and Reporting Conventions

For EFE-Time, each dataset is evolved under three random seeds; reported MASE, wQL, and MAE values are means and standard deviations across those runs. Transfer experiments reuse the frozen programs learned with Chronos-2 without re-evolving or tuning them for the target forecaster.

For EFE-Tab, TabArena provides three repetitions of three-fold splits for the medium-sized datasets. Using all repetitions with three EFE seeds would require 27 evolution/evaluation runs per dataset, so the main benchmark uses the three folds from the first official repetition. Baseline comparisons against CAAFE and LLM-FE use the same 100-iteration LLM-query budget as EFE-Tab. Frozen feature programs are then evaluated with TabPFN, LightGBM, and single decision trees.

## F.5. The List of Datasets Used

*Table 3.* EFE-Time datasets used in the evaluation.

| User label | Official GIFT-Eval name | Freq. | Pred. len. | # series (train) | Series length (min / median / max) | # test instances |
|---|---|---|---|---|---|---|
| CovidDeaths | covid_deaths | D | 30 | 266 | 152 / 152 / 152 | 266 |
| SolarHourly | solar/H | H | 48 | 137 | 7,800 / 7,800 / 7,800 | 2,603 |
| M4Yearly | m4_yearly | A | 6 | 22,974 | 7 / 23 / 272 | 22,974 |
| JenaWeather (H) | jena_weather/H | H | 48 | 21 (univariate explode) | 7,824 / 7,824 / 7,824 | 399 |
| SZ_TAXI_15T | SZ_TAXI/15T | 15min | 48 | 156 | 2,592 / 2,592 / 2,592 | 1,092 |
| bitbrains_storage/H | bitbrains_fast_storage/H | H | 48 | 2,500 (1,250 × {read, write}) | 577 / 577 / 577 | 5,000 |
| bitbrains_rnd/H | bitbrains_rnd/H | H | 48 | 1,000 (500 × {read, write}) | 576 / 576 / 576 | 2,000 |
| us_births_D | us_births/D | D | 30 | 1 | 6,675 / 6,675 / 6,675 | 20 |
| kdd_cup_2018_H | kdd_cup_2018_with_missing/H | H | 48 | 270 | 8,496 / 9,890 / 9,912 | 5,400 |
| restaurant | restaurant | D | 30 | 807 | 7 / 236 / 418 | 807 |

*Table 4.* The TabArena datasets used in the EFE-Tab evaluation.

| User label | TabArena Name | OpenEvolve ID | Rows | Feat. |
|---|---|---|---|---|
| Churn | churn | 363623 | 5,000 | 19 |
| HRAnalyticsJobChange | HR_Analytics_Job_Change_of_Data_Scientists | 363679 | 19,158 | 12 |
| E-CommereShippingData | E-CommereShippingData | 363632 | 10,999 | 10 |
| in_vehicle_coupon_recommendation | in_vehicle_coupon_recommendation | 363681 | 12,684 | 24 |
| online_shoppers_intention | online_shoppers_intention | 363691 | 12,330 | 17 |
| Bank_Customer_Churn | Bank_Customer_Churn | 363619 | 10,000 | 10 |
| BankMarketing | bank-marketing | 363618 | 45,211 | 13 |
| Diabetes | diabetes | 363629 | 768 | 8 |
| Fitness_Club | Fitness_Club | 363671 | 1,500 | 6 |

## F.6. Synthetic Smooth Exponential Time-Series Generator

We generate synthetic exponential-growth time series designed to isolate smooth multiplicative growth from transient spike effects.

For each series, let the sequence length be $L$ and define $T = L - 1$, with time index

$$t \in \{0, 1, \ldots, T\}.$$

We first sample a starting value $s$ and an overall growth ratio $r$ log-uniformly:

$$s \sim \text{LogUniform}(s_{\min}, s_{\max}),$$

$$r \sim \text{LogUniform}(r_{\min}, r_{\max}).$$

Equivalently,

$$\log s \sim \text{Uniform}(\log s_{\min}, \log s_{\max}),$$

$$\log r \sim \text{Uniform}(\log r_{\min}, \log r_{\max}).$$

The total log-growth over the full sequence is therefore

$$G = \log r.$$

The generator constructs a monotone trajectory in log space. Let

$$\eta_i = \text{SmoothNoise}(T, \sigma), \qquad i = 1, \dots, T,$$

where SmoothNoise denotes Gaussian white noise smoothed by convolution with a one-dimensional Gaussian kernel of width controlled by $\sigma$, then normalized to have approximately zero mean and unit root-mean-square magnitude. We convert this smooth noise into positive growth weights:

$$w_i = \exp(\gamma \eta_i),$$

where $\gamma$ controls the volatility of the growth rate. These weights are normalized so that the total log-growth is exactly $G$:

$$\Delta \ell_i = \frac{w_i}{\sum_{j=1}^{T} w_j} G.$$

The latent log-trend is then

$$\ell_0 = \log s,$$

$$\ell_t = \log s + \sum_{i=1}^{t} \Delta \ell_i, \qquad t = 1, \dots, T.$$

The corresponding latent exponential trend is

$$m_t = \exp(\ell_t).$$

Because each $\Delta \ell_i$ is positive, the latent trend is monotone increasing. In addition, the normalization enforces exact endpoint control:

$$m_0 = s,$$

$$m_T = sr.$$

For reference, we also define the clean log-linear exponential baseline

$$c_t = sr^{t/T},$$

which is the trajectory obtained by distributing the total log-growth uniformly over time.

Finally, the observable time series is obtained by applying smooth multiplicative observation noise. Let

$$\varepsilon_t = \alpha \, \text{SmoothNoise}(L, \sigma/2),$$

where $\alpha$ controls the observation noise scale. The final observed sequence is

$$x_t = m_t \exp(\varepsilon_t).$$

Equivalently,

$$x_t = \exp(\ell_t + \varepsilon_t).$$

Combining the above definitions, the generated series can be written as

$$x_t = \exp\left( \log s + \sum_{i=1}^{t} \frac{\exp(\gamma \eta_i)}{\sum_{j=1}^{T} \exp(\gamma \eta_j)} \log r + \varepsilon_t \right)$$

for $t = 0, \dots, T$, with the convention that the summation term is zero when $t = 0$.

In the experiments, the default parameter ranges are

$$s \sim \text{LogUniform}(0.5, 2.0),$$

$$r \sim \text{LogUniform}(5000, 8000),$$

$$\gamma \sim \text{Uniform}(0.2, 1.2),$$

$$\alpha \sim \text{Uniform}(0.0, 0.08),$$

$$\sigma \sim \text{Uniform}(8.0, 32.0).$$

Each time series draws its own independent values of $s$, $r$, $\gamma$, $\alpha$, and $\sigma$.

*Table 5.* Model-size scan of the EFE-Time evolution loop on `covid_deaths` and `solar_H`. All runs use 300 evolution iterations and three seeds; the reported program per seed is the optimizer's combined-score winner (frozen at `iteration_found`, evaluated on the full held-out test split). Each Chronos-2 zero-shot baseline is constant across sizes within the dataset block and is therefore reported once. The 4B model is served locally via vLLM (Kwon et al., 2023) on a single RTX A6000 (two A6000s for 27B and 35B-A3B); larger sizes are served through a hosted Qwen API endpoint.

| LLM | Baseline | | | EFE-Time | | | % Improvement | | |
|---|---|---|---|---|---|---|---|---|---|
| | MASE | WQL | MAE | MASE | WQL | MAE | MASE | WQL | MAE |
| *Dataset:* `covid_deaths` | | | | | | | | | |
| Qwen3.5-4B | | | | $34.116 \pm 2.334$ | $0.035 \pm 0.007$ | $111.051 \pm 22.272$ | $3.8 \pm 6.6$ | $10.4 \pm 17.8$ | $10.4 \pm 18.0$ |
| Qwen3.5-9B | 35.461 | 0.039 | 123.880 | $33.573 \pm 1.170$ | $0.159 \pm 0.102$ | $133.536 \pm 13.124$ | $5.3 \pm 3.3$ | $-301.8 \pm 258.2$ | $-7.8 \pm 10.6$ |
| Qwen3.5-27B | | | | $34.849 \pm 0.527$ | $0.040 \pm 0.002$ | $128.647 \pm 9.975$ | $1.7 \pm 1.5$ | $-1.9 \pm 5.1$ | $-3.8 \pm 8.1$ |
| Qwen3.5-35B-A3B | | | | $34.994 \pm 0.484$ | $0.153 \pm 0.105$ | $116.952 \pm 12.132$ | $1.3 \pm 1.4$ | $-288.9 \pm 267.0$ | $5.6 \pm 9.8$ |
| *Dataset:* `solar_H` | | | | | | | | | |
| Qwen3.5-4B | | | | $0.919 \pm 0.069$ | $0.334 \pm 0.011$ | $11.912 \pm 0.911$ | $7.8 \pm 6.9$ | $5.1 \pm 3.0$ | $7.9 \pm 7.0$ |
| Qwen3.5-9B | 0.996 | 0.352 | 12.936 | $0.885 \pm 0.073$ | $0.330 \pm 0.011$ | $11.473 \pm 0.975$ | $11.2 \pm 7.4$ | $6.1 \pm 3.0$ | $11.3 \pm 7.5$ |
| Qwen3.5-27B | | | | $0.962 \pm 0.024$ | $0.340 \pm 0.007$ | $12.496 \pm 0.315$ | $3.5 \pm 2.4$ | $3.2 \pm 2.1$ | $3.4 \pm 2.4$ |
| Qwen3.5-35B-A3B | | | | $0.887 \pm 0.072$ | $0.333 \pm 0.008$ | $11.489 \pm 0.961$ | $11.0 \pm 7.2$ | $5.3 \pm 2.4$ | $11.2 \pm 7.4$ |

## F.7. Compute

Except for running local LLMs detailed in Section F.8, we have only used 4 L40S GPUs throughout experiments, with 8 CPU cores and 128 GB of memory.

## F.8. Local-model evolution

The main paper used a hosted frontier LLM to drive evolution. To test whether the same loop also works with smaller open-weight models, we repeated the experiment on `covid_deaths` and `solar_H` using four Qwen3.5 sizes: 4B, 9B, 27B, and 35B-A3B. All non-model settings were held fixed. The 4B and 9B models were served locally with vLLM (Kwon et al., 2023) on one RTX A6000, while 27B and 35B-A3B used two A6000s or a hosted Qwen endpoint. Each setting was run for 300 iterations with three random seeds.

Table 5 reports the final optimizer-selected program for each setting. On `covid_deaths`, every model size found useful MASE improvements, but larger models were not consistently better. The 9B model had the largest mean MASE gain (+5.3 ± 3.3%), while 4B also improved and matched the larger models on its best seed. However, some configurations produced poorly calibrated quantiles: 9B and 35B-A3B both showed large WQL regressions, even when their median forecasts improved. The 27B model was more stable in WQL but had the smallest MASE gain. Overall, performance did not scale monotonically with model size.

The `solar_H` results were cleaner. All four sizes improved MASE, WQL, and MAE on average, with no calibration failures like those seen on `covid_deaths`. The 9B and 35B-A3B models performed best, reaching about +11% mean MASE improvement, while 4B still achieved a strong mean gain of +7.8 ± 6.9%. The 27B model was less strong but more consistent. Much of the variance came from whether a seed discovered a high-performing program: successful seeds reached a ~15% improvement regime, while weaker seeds stayed close to baseline. As before, gains did not follow a smooth model-size scaling trend.

Yet, in then end, our results demonstrate that end-to-end program evolution, with our existing framework and large context sizes provided to LLMs, is not stable for non-frontier, small-scale models.

## G. Extended Results

We present here our raw results in Table 6 and in Table 7. We also demonstrate EFE-Time evolutions in Figure 8 and Figure 9 across 3 repetitions.

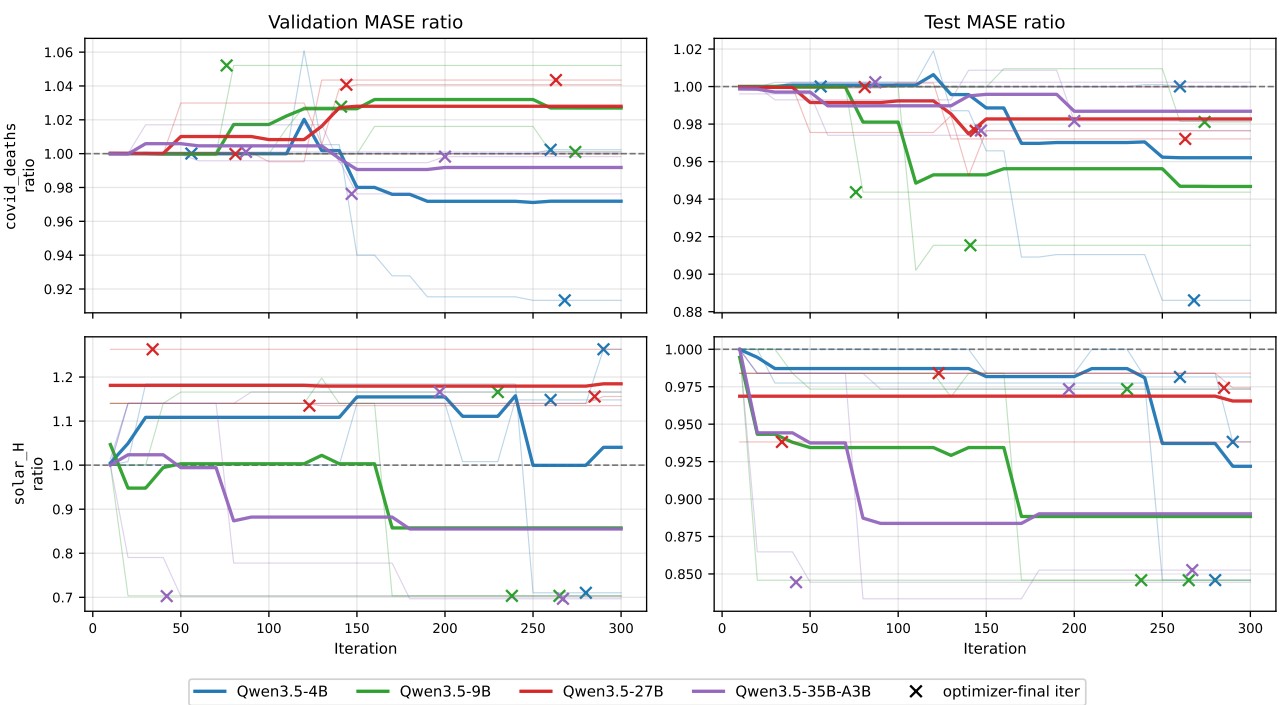

*Figure 7.* Per-checkpoint validation and test MASE ratios for the `covid_deaths` and `solar_H` model-size scans. Faint traces are individual seeds; solid lines are per-size means; ×-marks denote the iteration at which each run's optimizer-final program was first found. Values below the dashed line (ratio=1) improve over the Chronos-2 identity-transform baseline.

*Table 6.* Performance comparison of EFE-Time against identity transformation across time-series datasets. The EFE-Time transformations are learned for Chronos-2, and other models reuses them. As Reverso does not output quantiles, we take the $\mathrm{WQL}_{\mathrm{reverso}} ;=; \frac{\sum_t |\mathrm{actual}_t - \mathrm{median}_t|}{\sum_t |\mathrm{actual}_t|}$

| Model | Dataset | Baseline | | | EFE-Time | | | % Improvement | | |
|---|---|---|---|---|---|---|---|---|---|---|
| | | MASE | WQL | MAE | MASE | WQL | MAE | MASE | WQL | MAE |
| **Chronos-2** | CovidDeaths | 35.444 | 0.039 | 124.241 | 31.853 ± 0.835 | 0.032 ± 0.003 | 102.830 ± 13.968 | 10.1 ± 2.3 | 19.5 ± 8.7 | 17.2 ± 11.2 |
| | SolarHourly | 0.996 | 0.352 | 12.934 | 0.915 ± 0.049 | 0.335 ± 0.002 | 11.883 ± 0.648 | 8.1 ± 4.9 | 4.7 ± 0.7 | 8.1 ± 5.0 |
| | M4Yearly | 3.370 | 0.117 | 905.276 | 3.183 ± 0.135 | 0.115 ± 0.002 | 868.517 ± 26.300 | 5.6 ± 3.9 | 2.1 ± 1.6 | 4.1 ± 2.9 |
| | JenaWeather (H) | 0.547 | 0.042 | 8.745 | 0.538 ± 0.004 | 0.041 ± 0.001 | 8.436 ± 0.199 | 1.6 ± 0.6 | 2.1 ± 1.3 | 3.5 ± 2.3 |
| | SZ_TAXI_15T | 0.543 | 0.200 | 2.722 | 0.543 ± 0.000 | 0.200 ± 0.000 | 2.722 ± 0.001 | 0.0 ± 0.0 | 0.0 ± 0.0 | 0.0 ± 0.0 |
| | bitbrains_storage/H | 0.993 | 0.815 | 384.949 | 0.993 ± 0.000 | 0.815 ± 0.000 | 384.949 ± 0.000 | 0.0 ± 0.0 | 0.0 ± 0.0 | 0.0 ± 0.0 |
| | bitbrains_rnd/H | 5.821 | 1.032 | 175.961 | 5.818 ± 0.002 | 1.016 ± 0.022 | 168.439 ± 10.611 | 0.1 ± 0.0 | 4.3 ± 6.0 | 1.5 ± 2.1 |
| | us_births_D | 0.345 | 0.017 | 234.437 | 0.342 ± 0.002 | 0.017 ± 0.000 | 232.163 ± 1.191 | 1.0 ± 0.5 | 1.0 ± 0.5 | 0.4 ± 0.2 |
| | kdd_cup_2018_H | 0.999 | 0.396 | 24.712 | 0.971 ± 0.004 | 0.388 ± 0.002 | 24.281 ± 0.213 | 2.8 ± 0.4 | 2.0 ± 0.6 | 1.7 ± 0.9 |
| | restaurant | 0.684 | 0.256 | 7.055 | 0.682 ± 0.002 | 0.256 ± 0.001 | 7.041 ± 0.016 | 0.3 ± 0.2 | −0.2 ± 0.2 | 0.2 ± 0.2 |
| | **Mean % Improvement** | – | | | – | | | **3.0** | **3.6** | **3.7** |
| **TimesFM-2.5** | CovidDeaths | 36.909 | 0.035 | 112.492 | 34.049 ± 2.461 | 0.030 ± 0.004 | 96.942 ± 16.612 | 7.7 ± 6.6 | 13.8 ± 13.0 | 13.8 ± 14.7 |
| | SolarHourly | 0.912 | 0.348 | 11.832 | 0.857 ± 0.006 | 0.324 ± 0.006 | 11.112 ± 0.105 | 6.0 ± 0.7 | 6.1 ± 0.9 | 6.9 ± 1.9 |
| | M4Yearly | 3.575 | 0.127 | 953.276 | 3.288 ± 0.176 | 0.121 ± 0.004 | 896.327 ± 41.772 | 8.0 ± 4.9 | 6.0 ± 4.4 | 4.9 ± 3.4 |
| | JenaWeather (H) | 0.525 | 0.044 | 8.679 | 0.521 ± 0.001 | 0.041 ± 0.001 | 8.374 ± 0.139 | 0.9 ± 0.2 | 4.2 ± 2.1 | 3.5 ± 1.6 |
| | SZ_TAXI_15T | 0.563 | 0.136 | 1.849 | 0.561 ± 0.000 | 0.135 ± 0.000 | 1.844 ± 0.000 | 0.3 ± 0.0 | 0.2 ± 0.0 | 0.3 ± 0.0 |
| | bitbrains_storage/H | 1.109 | 0.791 | 370.809 | 1.125 ± 0.012 | 0.791 ± 0.000 | 370.702 ± 0.096 | −1.5 ± 1.1 | 0.0 ± 0.0 | 0.0 ± 0.0 |
| | bitbrains_rnd/H | 5.854 | 0.630 | 150.332 | 5.854 ± 0.000 | 0.630 ± 0.000 | 150.331 ± 0.001 | 0.0 ± 0.0 | 0.0 ± 0.0 | 0.0 ± 0.0 |
| | us_births_D | 0.338 | 0.018 | 229.720 | 0.338 ± 0.000 | 0.018 ± 0.000 | 229.720 ± 0.001 | 0.0 ± 0.0 | 0.0 ± 0.0 | 0.0 ± 0.0 |
| | kdd_cup_2018_H | 0.952 | 0.381 | 23.574 | 0.952 ± 0.000 | 0.381 ± 0.000 | 23.574 ± 0.000 | 0.0 ± 0.0 | 0.0 ± 0.0 | 0.0 ± 0.0 |
| | restaurant | 0.685 | 0.257 | 7.075 | 0.681 ± 0.003 | 0.256 ± 0.001 | 7.029 ± 0.034 | 0.6 ± 0.4 | 0.3 ± 0.2 | 0.7 ± 0.5 |
| | **Mean % Improvement** | – | | | – | | | **2.2** | **3.1** | **3.0** |
| **Moirai-2-Small** | CovidDeaths | 36.958 | 0.028 | 91.000 | 33.791 ± 0.128 | 0.026 ± 0.000 | 86.894 ± 1.278 | 8.6 ± 0.3 | 6.0 ± 1.6 | 4.5 ± 1.4 |
| | SolarHourly | 0.879 | 0.342 | 11.403 | 0.832 ± 0.004 | 0.314 ± 0.004 | 10.790 ± 0.049 | 5.4 ± 0.4 | 8.3 ± 1.0 | 5.4 ± 0.4 |
| | M4Yearly | 3.320 | 0.116 | 890.298 | 3.189 ± 0.151 | 0.117 ± 0.003 | 870.904 ± 30.403 | 4.0 ± 4.5 | −1.0 ± 2.5 | 2.2 ± 3.4 |
| | JenaWeather (H) | 0.536 | 0.042 | 8.454 | 0.533 ± 0.001 | 0.041 ± 0.000 | 8.304 ± 0.066 | 0.6 ± 0.2 | 1.6 ± 0.7 | 1.8 ± 0.8 |
| | SZ_TAXI_15T | 0.546 | 0.201 | 2.739 | 0.546 ± 0.000 | 0.201 ± 0.000 | 2.739 ± 0.000 | 0.0 ± 0.0 | 0.0 ± 0.0 | 0.0 ± 0.0 |
| | bitbrains_storage/H | 1.128 | 0.615 | 291.618 | 1.136 ± 0.006 | 0.615 ± 0.000 | 291.388 ± 0.305 | −0.7 ± 0.5 | 0.0 ± 0.0 | 0.1 ± 0.1 |
| | bitbrains_rnd/H | 5.809 | 0.670 | 195.729 | 5.841 ± 0.018 | 0.673 ± 0.016 | 192.433 ± 9.768 | −0.5 ± 0.3 | −0.5 ± 2.4 | 1.7 ± 5.0 |
| | us_births_D | 0.373 | 0.020 | 253.573 | 0.369 ± 0.006 | 0.019 ± 0.000 | 250.843 ± 3.861 | 1.1 ± 1.5 | 0.8 ± 1.1 | 1.1 ± 1.5 |
| | kdd_cup_2018_H | 1.012 | 0.426 | 26.328 | 0.988 ± 0.005 | 0.415 ± 0.004 | 25.173 ± 0.114 | 2.4 ± 0.5 | 2.6 ± 1.0 | 4.4 ± 0.4 |
| | restaurant | 0.702 | 0.260 | 7.196 | 0.691 ± 0.012 | 0.261 ± 0.005 | 7.104 ± 0.077 | 1.6 ± 1.7 | −0.4 ± 2.1 | 1.3 ± 1.1 |
| | **Mean % Improvement** | – | | | – | | | **2.2** | **1.7** | **2.2** |
| **Reverso-Nano** | CovidDeaths | 43.852 | 0.105 | 278.727 | 39.202 ± 2.648 | 0.076 ± 0.014 | 202.023 ± 36.430 | 10.6 ± 6.0 | 27.5 ± 13.1 | 27.5 ± 13.1 |
| | SolarHourly | 0.874 | 0.416 | 11.299 | 0.870 ± 0.005 | 0.415 ± 0.002 | 11.258 ± 0.058 | 0.4 ± 0.5 | 0.4 ± 0.5 | 0.4 ± 0.5 |
| | M4Yearly | 3.480 | 0.148 | 920.348 | 3.147 ± 0.066 | 0.138 ± 0.002 | 861.605 ± 10.120 | 9.6 ± 1.9 | 6.4 ± 1.1 | 6.4 ± 1.1 |
| | JenaWeather (H) | 0.538 | 0.051 | 8.313 | 0.534 ± 0.001 | 0.050 ± 0.000 | 8.183 ± 0.019 | 0.8 ± 0.1 | 1.6 ± 0.2 | 1.6 ± 0.2 |
| | SZ_TAXI_15T | 0.556 | 0.260 | 2.781 | 0.550 ± 0.000 | 0.258 ± 0.000 | 2.755 ± 0.000 | 1.0 ± 0.0 | 0.9 ± 0.0 | 0.9 ± 0.0 |
| | bitbrains_storage/H | 1.109 | 1.029 | 360.326 | 1.115 ± 0.009 | 1.028 ± 0.000 | 360.204 ± 0.172 | −0.6 ± 0.8 | 0.0 ± 0.0 | 0.0 ± 0.0 |
| | bitbrains_rnd/H | 5.857 | 0.734 | 166.715 | 5.857 ± 0.000 | 0.734 ± 0.000 | 166.714 ± 0.002 | 0.0 ± 0.0 | 0.0 ± 0.0 | 0.0 ± 0.0 |
| | us_births_D | 0.394 | 0.025 | 267.837 | 0.393 ± 0.001 | 0.025 ± 0.000 | 267.325 ± 0.724 | 0.2 ± 0.3 | 0.2 ± 0.3 | 0.2 ± 0.3 |
| | kdd_cup_2018_H | 0.978 | 0.493 | 24.488 | 0.978 ± 0.000 | 0.493 ± 0.000 | 24.488 ± 0.000 | 0.0 ± 0.0 | 0.0 ± 0.0 | 0.0 ± 0.0 |
| | restaurant | 0.713 | 0.341 | 7.381 | 0.697 ± 0.010 | 0.333 ± 0.006 | 7.204 ± 0.118 | 2.2 ± 1.4 | 2.3 ± 1.6 | 2.4 ± 1.6 |
| | **Mean % Improvement** | – | | | – | | | **2.4** | **3.9** | **3.9** |

*Table 7.* Performance comparison of feature engineering methods across tabular datasets over ROC-AUC. The reported results are 3 outer-fold averages from the TabArena benchmark.

| Model | Dataset | Feature Engineering Method | | | |
|---|---|---|---|---|---|
| | | **No FE** | **CAAFE** | **LLM-FE** | **EFE-Tab** |
| **Decision Tree** | Churn | .8944 ± .0178 | .9239 ± .0181 | .9080 ± .0213 | **.9266 ± .0108** |
| | DataScientists | .7898 ± .0069 | .7864 ± .0093 | .7914 ± .0065 | **.7923 ± .0041** |
| | CommereShipng | .7417 ± .0037 | .7441 ± .0034 | .7433 ± .0032 | **.7448 ± .0035** |
| | CouponRec | .7296 ± .0140 | .7285 ± .0124 | .6720 ± .1042 | **.7473 ± .0134** |
| | OnlineShoppers | .9235 ± .0068 | .9242 ± .0041 | **.9382 ± .0133** | .9235 ± .0040 |
| | BankCustomers | .8340 ± .0035 | .8409 ± .0077 | **.8502 ± .0096** | .8434 ± .0040 |
| | BankMarketing | .7329 ± .0056 | .7339 ± .0055 | .7345 ± .0048 | **.7348 ± .0064** |
| | Diabetes | .7974 ± .0185 | .7852 ± .0473 | .7909 ± .0112 | **.8136 ± .0082** |
| | FitnessClub | .8042 ± .0118 | .7943 ± .0071 | .7971 ± .0105 | **.8048 ± .0128** |
| | **Mean Rank** | 3.17 | 3.00 | 2.44 | **1.39** |
| **LightGBM** | Churn | .9251 ± .0136 | **.9305 ± .0076** | .9251 ± .0082 | .9302 ± .0097 |
| | DataScientists | .8046 ± .0069 | .8034 ± .0088 | .8027 ± .0063 | **.8052 ± .0073** |
| | CommereShipng | .7400 ± .0043 | .7437 ± .0039 | .7417 ± .0056 | **.7465 ± .0031** |
| | CouponRec | .8224 ± .0089 | .8223 ± .0073 | .8026 ± .0395 | **.8293 ± .0077** |
| | OnlineShoppers | .9318 ± .0056 | .9339 ± .0035 | **.9427 ± .0137** | .9342 ± .0035 |
| | BankCustomers | .8646 ± .0055 | .8709 ± .0088 | **.8938 ± .0070** | .8674 ± .0064 |
| | BankMarketing | .7636 ± .0067 | .7634 ± .0062 | **.7643 ± .0074** | .7634 ± .0080 |
| | Diabetes | .8318 ± .0136 | .8200 ± .0105 | **.8352 ± .0123** | .8297 ± .0148 |
| | FitnessClub | .8081 ± .0081 | .8029 ± .0009 | .8023 ± .0047 | **.8150 ± .0119** |
| | **Mean Rank** | 2.83 | 2.72 | 2.50 | **1.94** |
| **TabPFN** | Churn | .9307 ± .0087 | .9317 ± .0090 | .9331 ± .0063 | **.9338 ± .0068** |
| | DataScientists | .8040 ± .0057 | .8037 ± .0063 | .7977 ± .0051 | **.8042 ± .0059** |
| | CommereShipng | .7457 ± .0021 | **.7465 ± .0020** | .7423 ± .0019 | .7452 ± .0029 |
| | CouponRec | .8406 ± .0078 | **.8431 ± .0064** | .8006 ± .0597 | .8430 ± .0086 |
| | OnlineShoppers | .9360 ± .0044 | .9371 ± .0038 | **.9461 ± .0112** | .9376 ± .0037 |
| | BankCustomers | .8730 ± .0080 | .8751 ± .0090 | **.8975 ± .0060** | .8745 ± .0086 |
| | BankMarketing | .7619 ± .0071 | .7620 ± .0072 | .7597 ± .0038 | **.7625 ± .0079** |
| | Diabetes | **.8478 ± .0029** | .8430 ± .0059 | .8451 ± .0076 | .8435 ± .0056 |
| | FitnessClub | .8216 ± .0102 | .8162 ± .0067 | **.8219 ± .0096** | .8132 ± .0122 |
| | **Mean Rank** | 2.78 | 2.44 | 2.56 | **2.22** |

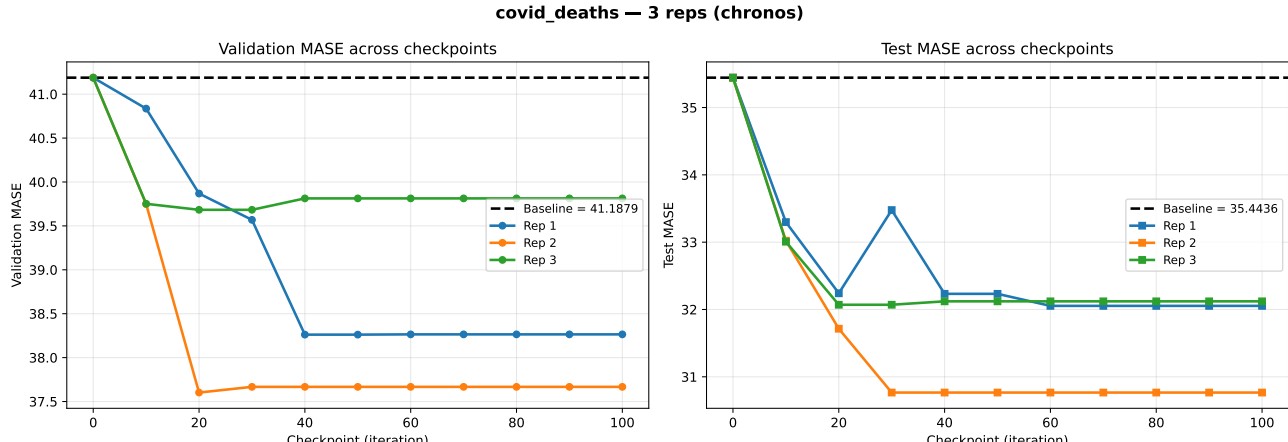

*Figure 8.* The change of validation and test MASE during EFE-Time evolution for Covid-Deaths.

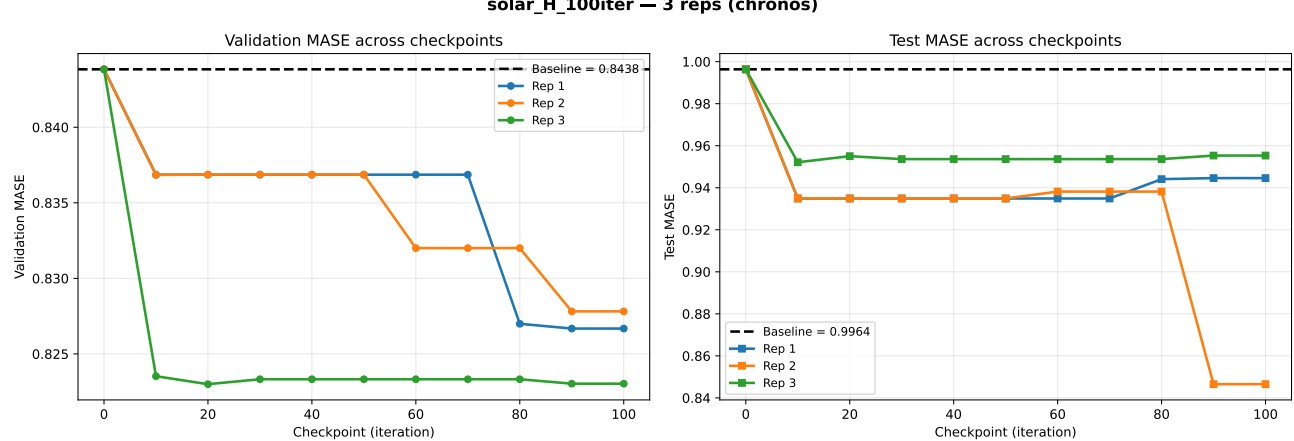

*Figure 9.* The change of validation and test MASE during EFE-Time evolution for Solar-hourly.

# H. Per-series forecast trajectories under the evolved transform

This section visualizes the per-series effect of the evolved transform on a small selection of test instances drawn from each dataset in the EFE-Time evaluation suite. Each three-panel column shows, top to bottom, Chronos-2 on the raw input window, Chronos-2 in the transformed space (the forecaster sees $z = \texttt{transform}(y)$ and predicts the transformed continuation), and the inverse-transformed forecast plotted back on the original scale. The four picks per dataset include both series where the transform helps and series where it has little or no effect, so the panels should be read as qualitative inspection rather than as a ranking. On `covid_deaths` and `solar_H` the transformed input window often resembles a near-periodic shape that Chronos-2 extrapolates more cleanly. We point this out in the per-dataset paragraphs where it applies. A recurring theme across the harder cases is the role of exogenous, perhaps unpredictable drivers with existing information: when the dynamics that govern the held-out window are not visible in the input history, no input-only transform can recover what is missing.

### H.1. `covid_deaths`

The transformed input on Fig. 10a and Fig. 10b is closer to a smooth periodic signal than the raw counter is, and Chronos-2's median forecast in the transformed space follows the gentle wave the model is now seeing. The inverse step then restores the original level and brings the forecast median into the neighbourhood of the held-out trajectory. Fig. 10c and Fig. 10d display sudden spikes in the held-out window that have no precedent in the input history: a death surge of this kind is typically triggered by an exogenous event (a regional outbreak, a hospital admission wave, a reporting catch-up). The dynamics governing such jumps are non-Markovian with respect to the model's input, which places a fundamental limit on what an input-only transform can recover. The transform absorbs some of the level shift and reduces the inverted-forecast MAE on both panels, but the residual on Fig. 10d widens the quantile cone enough that wQL regresses even as the median improves.

### H.2. `m4_yearly`

The M4-yearly horizon is short and the input windows are short as well. The transformed window typically rescales the trajectory's growth so that Chronos-2 in the transformed space sits closer to the ground truth's slope, and the inverse step then amplifies that corrected slope back to the original units as seen in Figure 11. We do not see the periodic transform-space shape that appears on `covid_deaths` and `solar_H`; the effect here is closer to a learned per-series scale and trend correction. The same caveat as on `covid_deaths` applies: the M4-yearly aggregates contain a substantial financial and macroeconomic component, and multi-year drift in those series is often driven by exogenous events (policy regimes, market shocks, demographic transitions) that the input window does not encode. Fig. 11d is an example where the transform tightens the slope but the held-out trajectory continues a regime change that no input-only mapping could anticipate, leaving a residual MASE above 16.

### H.3. `solar_H`

The hourly solar series are dominated by their diurnal cycle. In the transformed space as seen in Figure 12 the daily oscillation is regularized into a smoother periodic shape, and Chronos-2 follows the periodic continuation more closely than it does on the raw window; the inverse-transformed forecast then sits within a tight band around the held-out trajectory. Fig. 12d shows the opposite case: a short, near-flat segment where the transform shifts Chronos-2 off the true level by a small amount and every metric regresses. Solar irradiance is largely driven by deterministic astronomy plus weather, so the transform's leverage here is geometric rather than informational, and the failure case on Fig. 12d is a small one.

### H.4. `restaurant`

The transformed input window resembles the raw input on each of these picks, and the top and middle panel forecasts overlap closely as seen in Figure 13. The inverse-transformed median is essentially indistinguishable from the raw median, and the per-series gains in MASE, wQL, and MAE are small. The corpus is built from daily visitor counts at Japanese restaurants drawn from two reservation platforms, and the official held-out window spans the Golden Week holiday; days on which a restaurant was closed are excluded from scoring. The drivers of variation are therefore primarily exogenous: the holiday calendar, day-of-week effects modulated by closure decisions, and platform-level demand. None of these are encoded in the input history at the level of detail needed to anticipate the held-out trajectory, and an input-only transform cannot manufacture the missing signal. The aggregate near-identity behaviour reported in the main paper is consistent with this: when the relevant covariates live outside the history window, the optimizer settles close to the identity rather than

overfit a spurious mapping.

### H.5. `bitbrains_rnd/H`

The corpus is a workload trace from a virtualized datacenter hosting business-critical applications. The channels record requested and used resource demand on a heterogeneous shared cluster. Demand on such a cluster is driven mostly by external user activity and operational events (deployments, batch windows, traffic incidents), which makes the dynamics highly non-Markovian with respect to the channel-only input window we feed Chronos-2. The per-series picture is correspondingly heavy-tailed: Fig. 14c is a series where the transform compresses a high-magnitude spike into a smoother envelope and the inverse-transformed forecast tracks the held-out trajectory inside that envelope, dropping per-series MASE by roughly an order of magnitude. The other three picks retain the raw window's spiky character through the transform, and the inverse-transformed median sits close to the raw median. The aggregate near-identity behaviour reported in the main paper is therefore a mean over a heavy-tailed distribution of per-series effects, not a uniform absence of effect, and we do not extend the periodic transform-space observation from `covid_deaths` and `solar_H` to this dataset.

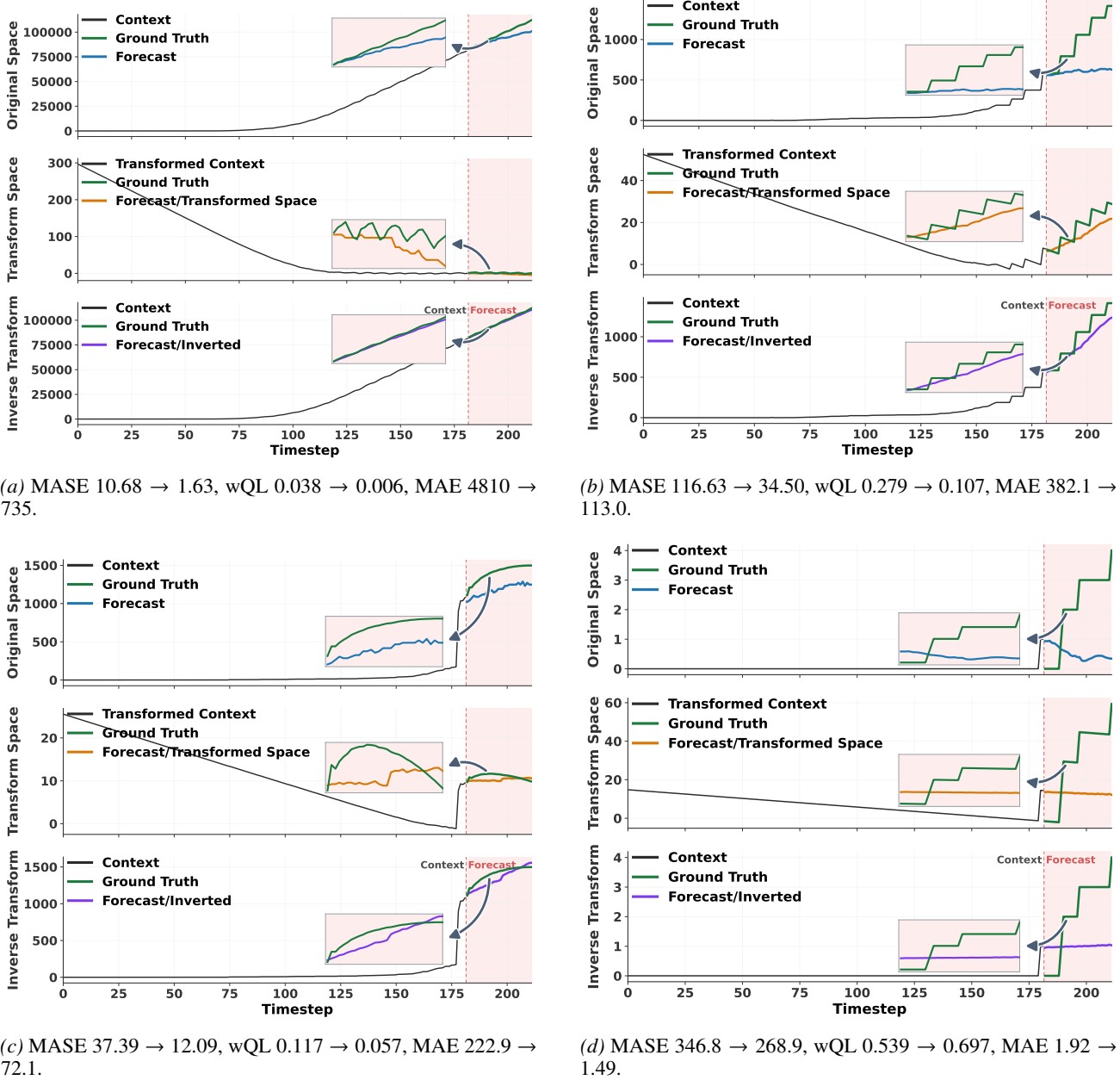

*(a)* MASE 10.68 → 1.63, wQL 0.038 → 0.006, MAE 4810 → 735.

*(b)* MASE 116.63 → 34.50, wQL 0.279 → 0.107, MAE 382.1 → 113.0.

*(c)* MASE 37.39 → 12.09, wQL 0.117 → 0.057, MAE 222.9 → 72.1.

*(d)* MASE 346.8 → 268.9, wQL 0.539 → 0.697, MAE 1.92 → 1.49.

*Figure 10.* Forecast trajectories on four `covid_deaths` test instances. Each three-panel column shows, top to bottom, Chronos-2 on the raw input, Chronos-2 in the transformed space, and the inverse-transformed forecast back on the original scale. Panels (a) and (b) show series where the transform substantially reduces the inverted-forecast error. Panels (c) and (d) show series whose held-out windows contain sudden spikes; the transform partially recovers the level on (c) and improves the median MAE but not the quantile loss on (d).

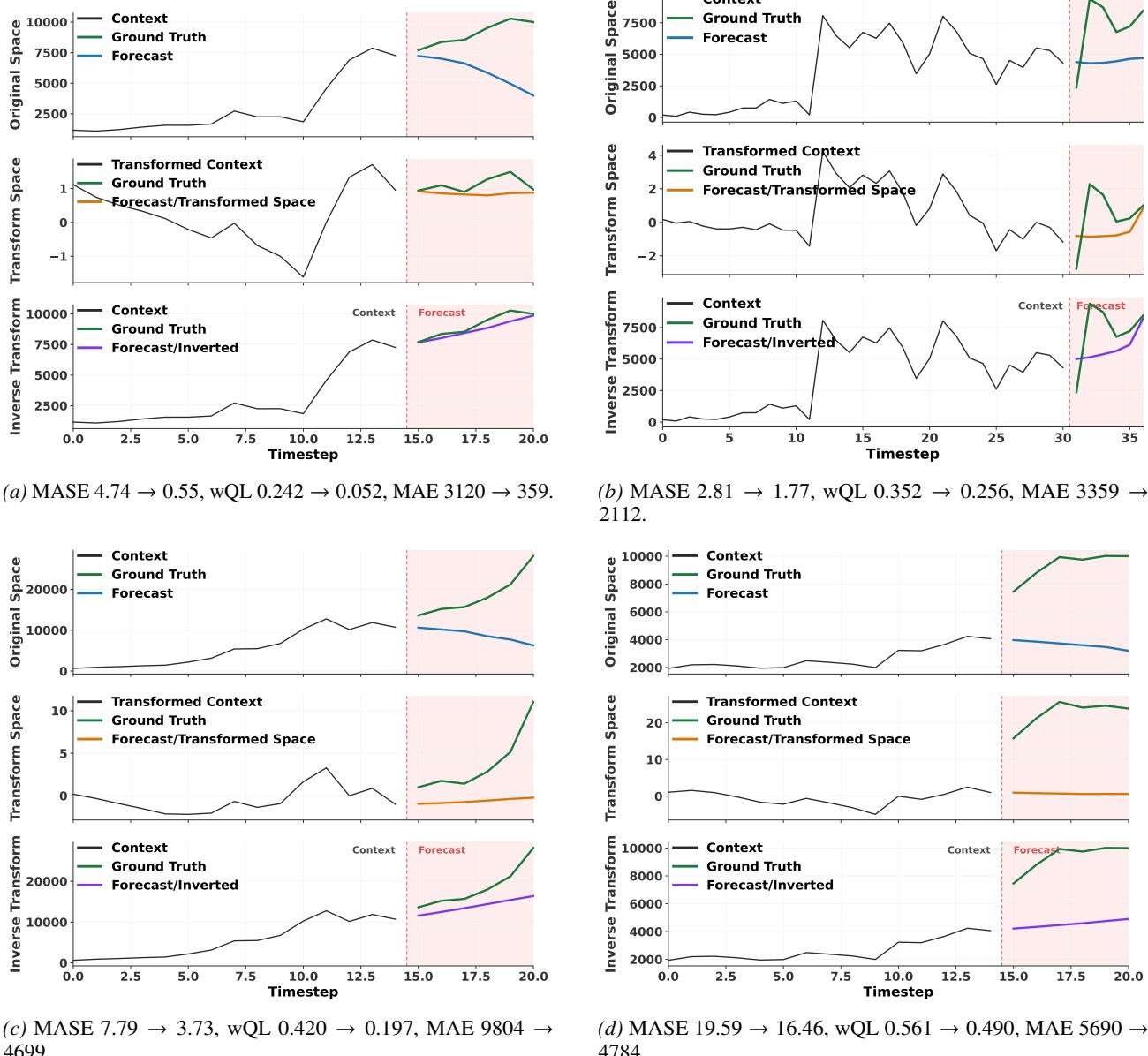

*(a)* MASE 4.74 → 0.55, wQL 0.242 → 0.052, MAE 3120 → 359.

*(b)* MASE 2.81 → 1.77, wQL 0.352 → 0.256, MAE 3359 → 2112.

*(c)* MASE 7.79 → 3.73, wQL 0.420 → 0.197, MAE 9804 → 4699.

*(d)* MASE 19.59 → 16.46, wQL 0.561 → 0.490, MAE 5690 → 4784.

*Figure 11.* Forecast trajectories on four `m4_yearly` test instances. The yearly horizon is six steps; the panels read as a per-series scale and trend correction rather than the periodic-shape rewriting visible on `covid_deaths` and `solar_H`. The forecast-horizon zoom inset is omitted because the six-step horizon is short enough to read directly from the panels.

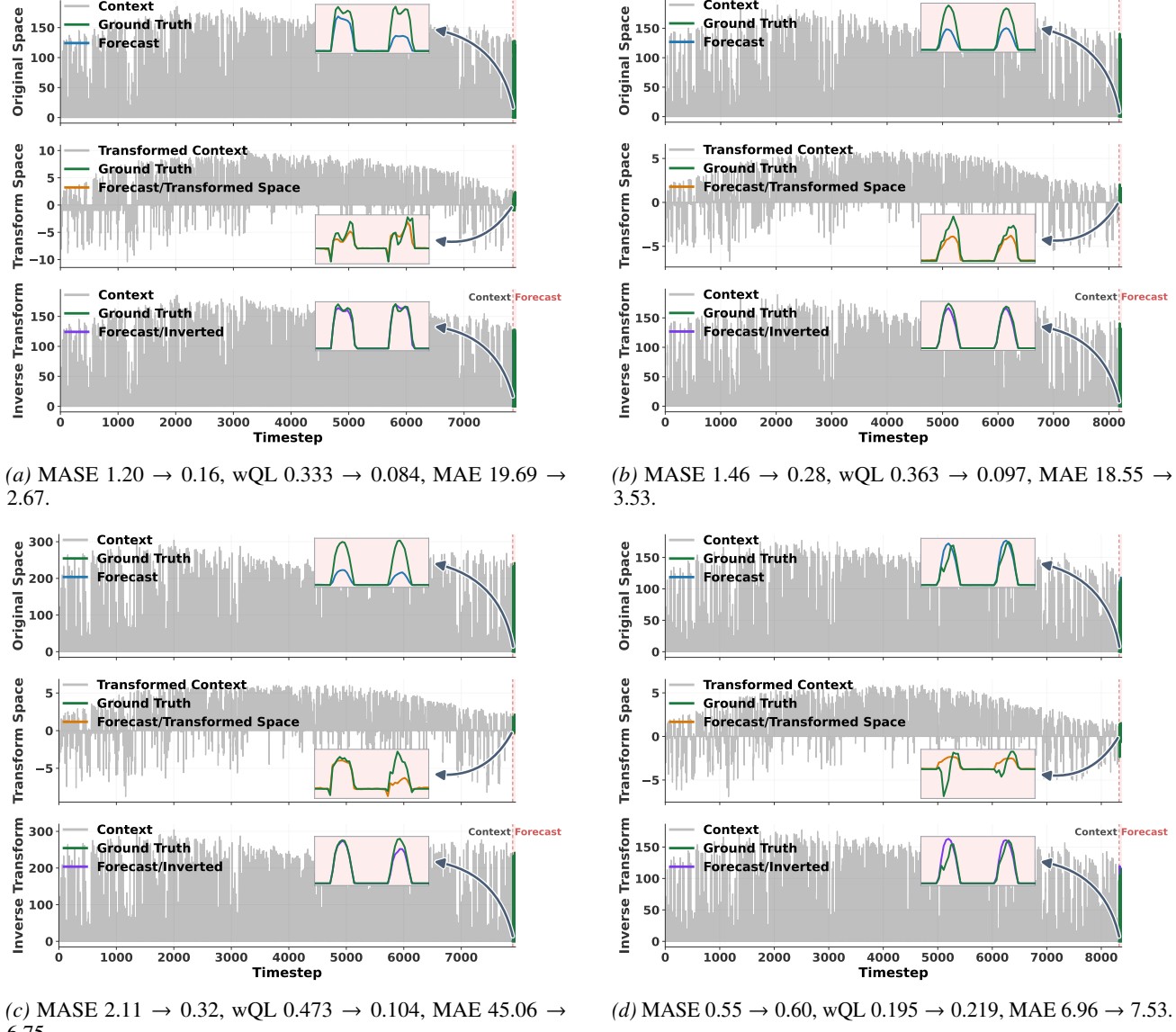

*(a)* MASE 1.20 → 0.16, wQL 0.333 → 0.084, MAE 19.69 → 2.67.

*(b)* MASE 1.46 → 0.28, wQL 0.363 → 0.097, MAE 18.55 → 3.53.

*(c)* MASE 2.11 → 0.32, wQL 0.473 → 0.104, MAE 45.06 → 6.75.

*(d)* MASE 0.55 → 0.60, wQL 0.195 → 0.219, MAE 6.96 → 7.53.

*Figure 12.* Forecast trajectories on four `solar_H` test instances. Hourly solar irradiance is strongly diurnal; on panels (a)-(c) the transform folds the daily cycle into a smoother oscillation that Chronos-2 extrapolates more cleanly. Panel (d) is included as a series where the transform leaves Chronos-2 slightly worse on every metric. The history line is rendered with reduced opacity so the forecast traces and the forecast-horizon inset stay legible against the long, densely diurnal context.

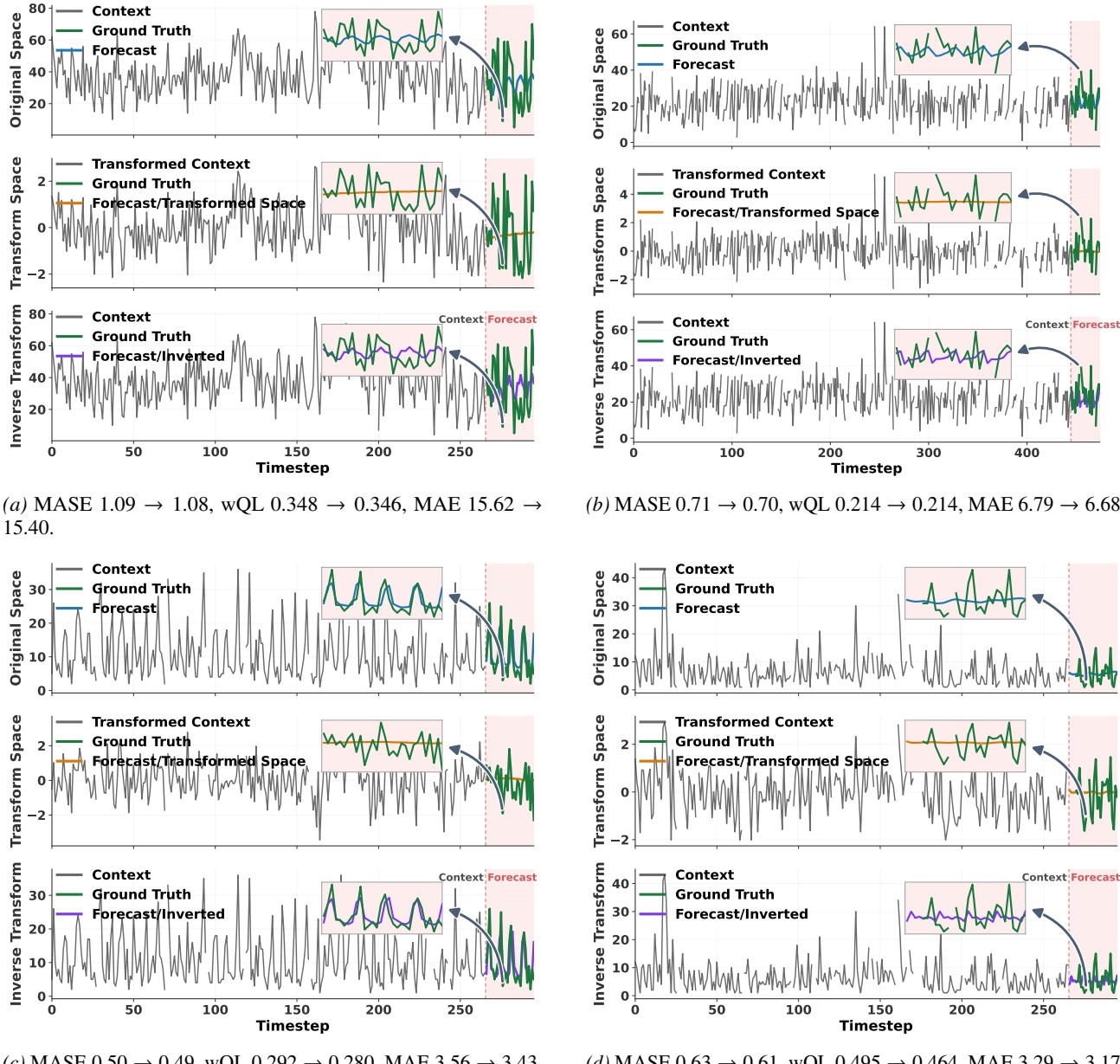

*(a)* MASE 1.09 → 1.08, wQL 0.348 → 0.346, MAE 15.62 → 15.40.

*(b)* MASE 0.71 → 0.70, wQL 0.214 → 0.214, MAE 6.79 → 6.68.

*(c)* MASE 0.50 → 0.49, wQL 0.292 → 0.280, MAE 3.56 → 3.43.

*(d)* MASE 0.63 → 0.61, wQL 0.495 → 0.464, MAE 3.29 → 3.17.

*Figure 13.* Forecast trajectories on four `restaurant` test instances. The dataset-aggregate test MASE shifts by roughly 0.6%; the per-series traces match this near-identity regime, with the transformed window resembling the raw window and the inverse-transformed median essentially indistinguishable from the raw median.

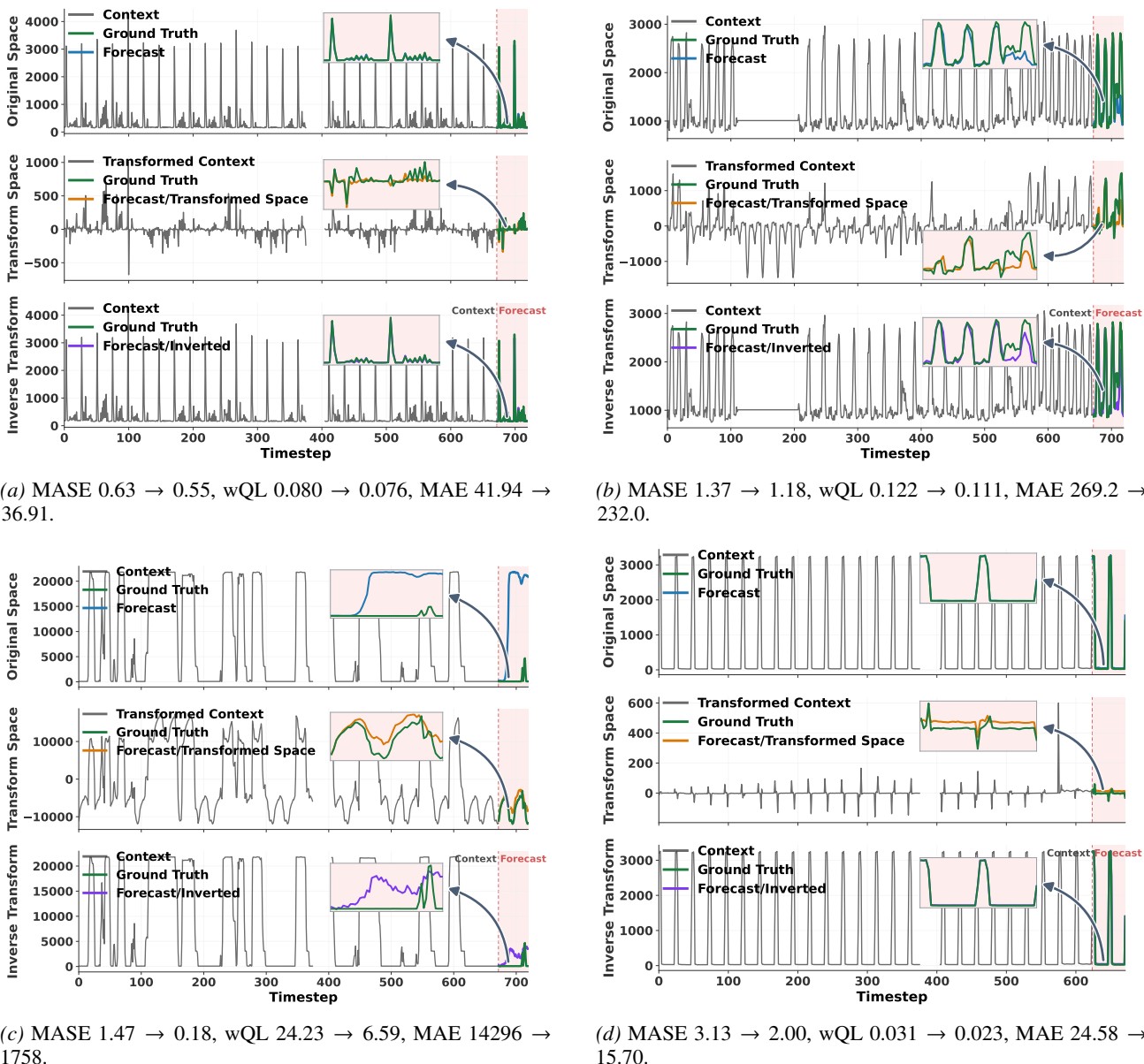

*(a)* MASE 0.63 → 0.55, wQL 0.080 → 0.076, MAE 41.94 → 36.91.

*(b)* MASE 1.37 → 1.18, wQL 0.122 → 0.111, MAE 269.2 → 232.0.

*(c)* MASE 1.47 → 0.18, wQL 24.23 → 6.59, MAE 14296 → 1758.

*(d)* MASE 3.13 → 2.00, wQL 0.031 → 0.023, MAE 24.58 → 15.70.

*Figure 14.* Forecast trajectories on four `bitbrains_rnd`/H test instances; all four are channel 0 of the bivariate input. The dataset-aggregate test MASE shifts by about 0.1%, but the per-series effect varies sharply: panel (c) shows an order-of-magnitude reduction in MASE on a single series, while panels (a), (b), and (d) show only modest gains and the transformed window retains the spiky character of the raw input.

# I. Evolution-tree visualization of the search process

This section visualizes the OpenEvolve search itself across the EFE-Time evaluation suite. Each evolution run is rendered as a single panel; the three random seeds for a given dataset are laid out horizontally so that within-dataset variability is visible at a glance. The intent is qualitative.

Every run is initialized from an identity-transform seed program at iteration 0 and proceeds for 100 iterations. At each iteration, the controller samples a parent program from the in-memory MAP-Elites database (three islands with periodic migration), prompts an LLM with the parent code together with a small set of inspiration and diversity exemplars, and asks for a complete rewrite. The new candidate is evaluated on the per-dataset evolution pool by running Chronos-2 zero-shot on the transformed history, inverse-transforming the forecast, and scoring the result against the identity baseline.

In every panel, each program is plotted at its (combined score, iteration) coordinates, coloured by the island the program was placed in. Faint grey segments are parent-to-child edges read from the per-program `parent_id` pointer in the run's saved checkpoints. The violet trace highlights the lineage of the run's final selection from root to leaf, and the violet star marks that program. The dashed grey line is the running best-so-far. The y-axis grows downward so the seed program sits at the top of every panel. A few panels show the violet trace truncated, or only the final star with no incoming line. This reflects the bounded MAP-Elites store rather than the search itself: as new candidates arrive, weaker programs are evicted from their feature cells, and a saved checkpoint persists only the active population at the moment of saving. A program that lives transiently in the database and then evicts before the next checkpoint write is never serialized to disk, so the parent JSON needed to draw an edge is missing. The lineage line therefore stops at the last ancestor that remained in the population.

The five panels reproduce, at the search-trajectory level, the qualitative pattern visible at the per-series level. On `covid_deaths` (Fig. 15) all three seeds locate strongly-improving programs that cross the 10%-improvement line; the seed-2 panel shows a long-lived ancestor whose lineage walks back to the seed, whereas the other two panels render only the final star because the winners' direct parents were evicted before the next checkpoint write. On `m4_yearly` (Fig. 16) the seed-to-seed variance is visibly wider, and only one seed crosses the 10%-improvement line; the slowest run flattens its best-so-far frontier in the first half of the budget and does not escape the corresponding plateau, which we read as the search getting stuck in a basin reachable by the LLM's rewrites rather than as evidence that no better program exists. On `solar_H` (Fig. 17) the three runs converge to similar combined scores in a narrow band; the populations crowd a tight vertical strip and the best-so-far frontier moves in small increments, consistent with the per-series picture from Sec. H.3, where the diurnal regularity of hourly solar irradiance gives Chronos-2 a strong baseline that bounds the headroom for any single transform. On `restaurant` (Fig. 18) and `bitbrains_rnd`/H (Fig. 19) the optimizer-selected scores sit just above the identity baseline; the populations crowd a near-vertical band around the identity score, the best-so-far frontiers inch rather than jump, and on `restaurant` one seed locks in its final selection in the early iterations and does not move thereafter. Read together with Sec. H.4 and Sec. H.5, this matches a search that does not find a transform with a meaningful effect on these corpora: the population drifts around identity across all three seeds, ruling out single-run failure as the explanation.

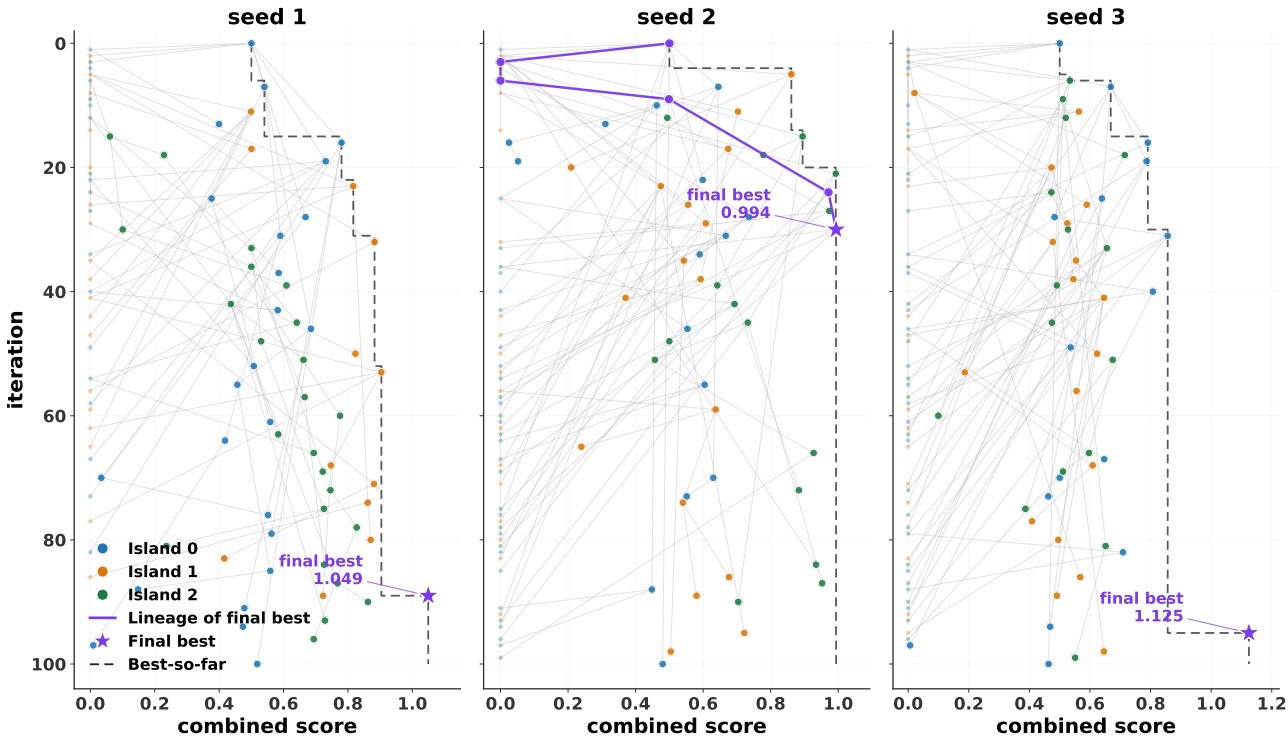

*Figure 15.* Evolution trees for the three `covid_deaths` runs (seeds 1, 2, 3 from left to right). All three seeds locate programs that cross the 10%-improvement line.

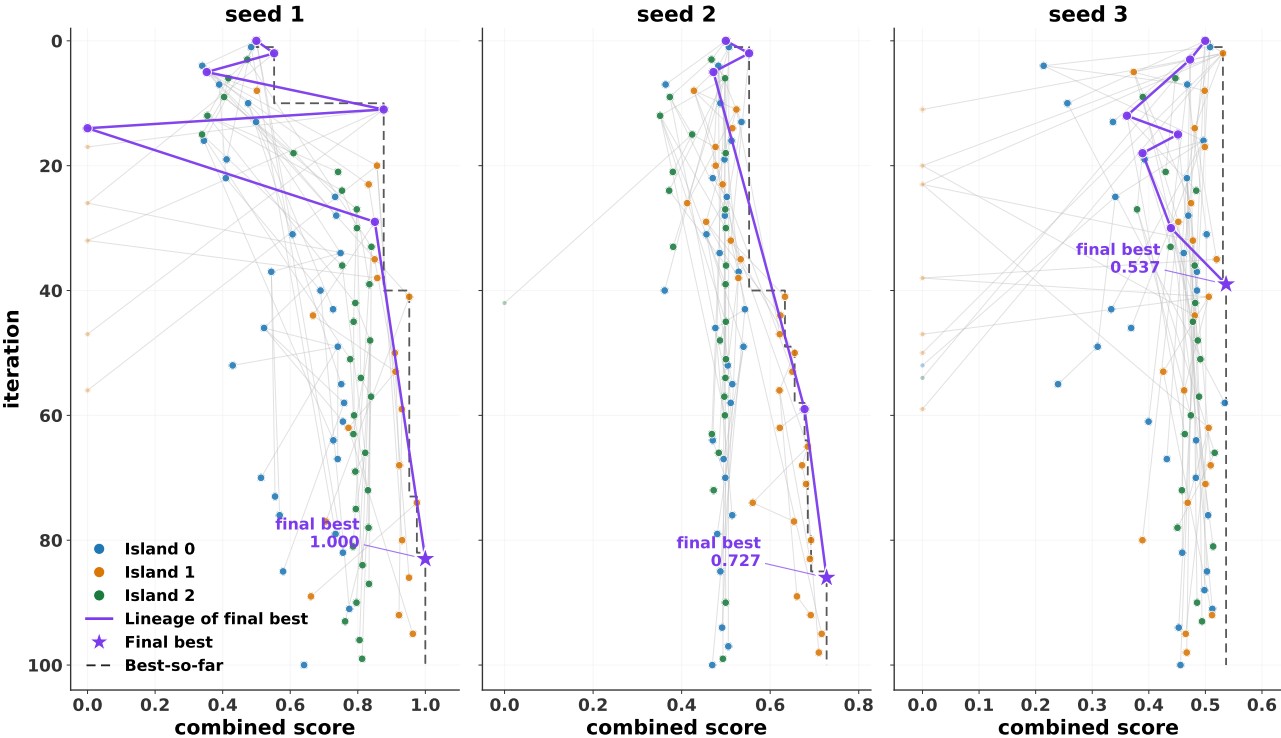

*Figure 16.* Evolution trees for the three `m4_yearly` runs. The seed-to-seed score variability is visibly wider than on `covid_deaths`.

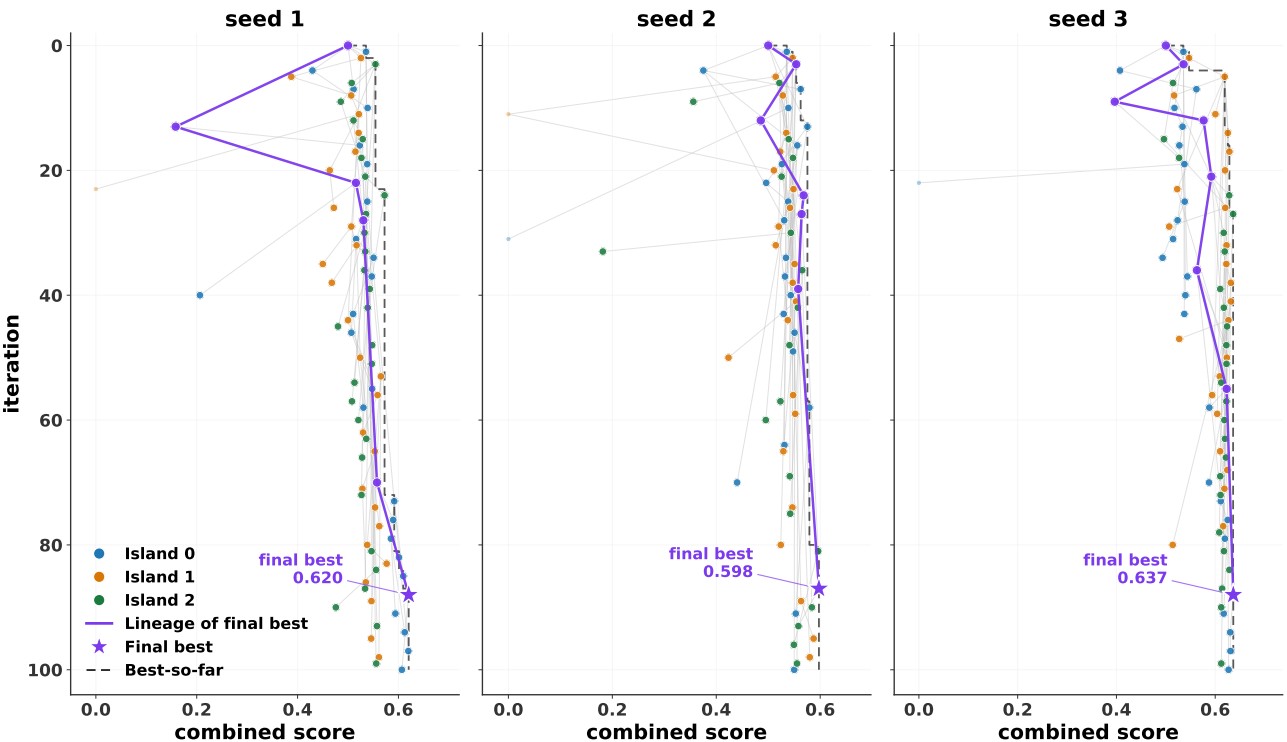

*Figure 17.* Evolution trees for the three solar_H runs. The three panels are visually similar.

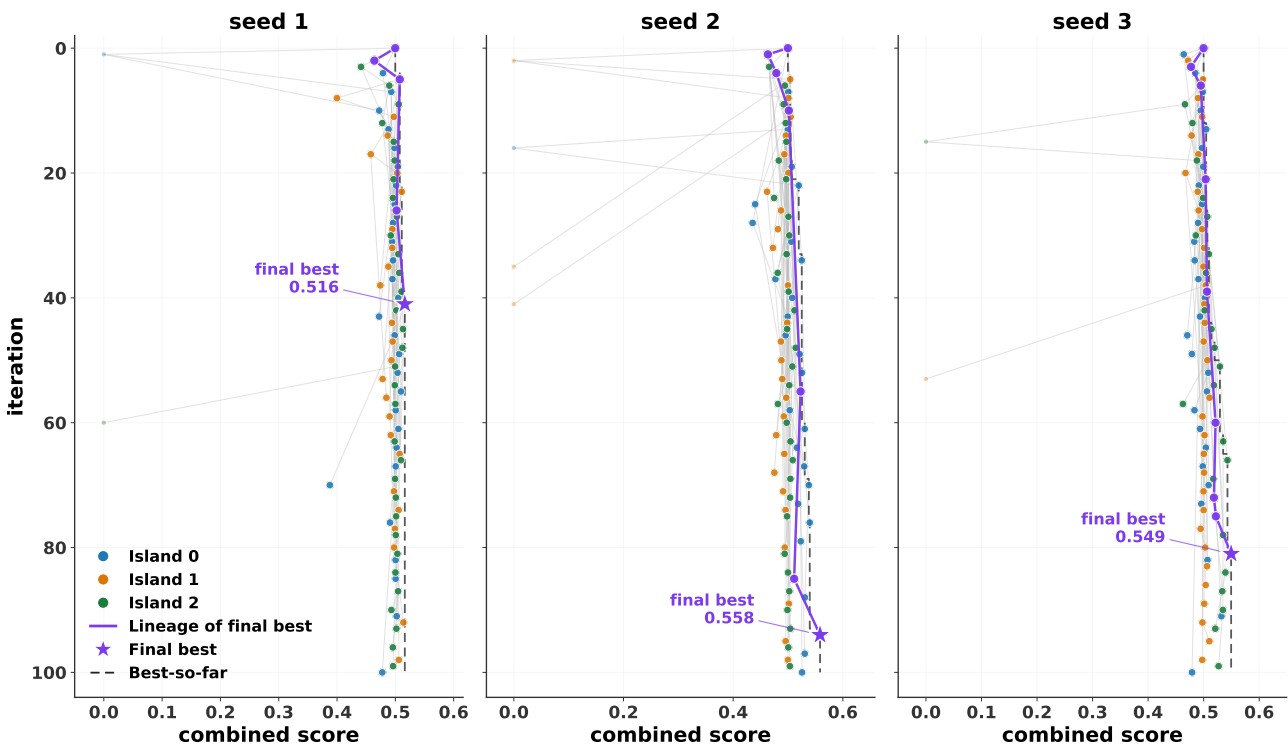

*Figure 18.* Evolution trees for the three restaurant runs. The optimizer-selected scores sit only marginally above the identity baseline.

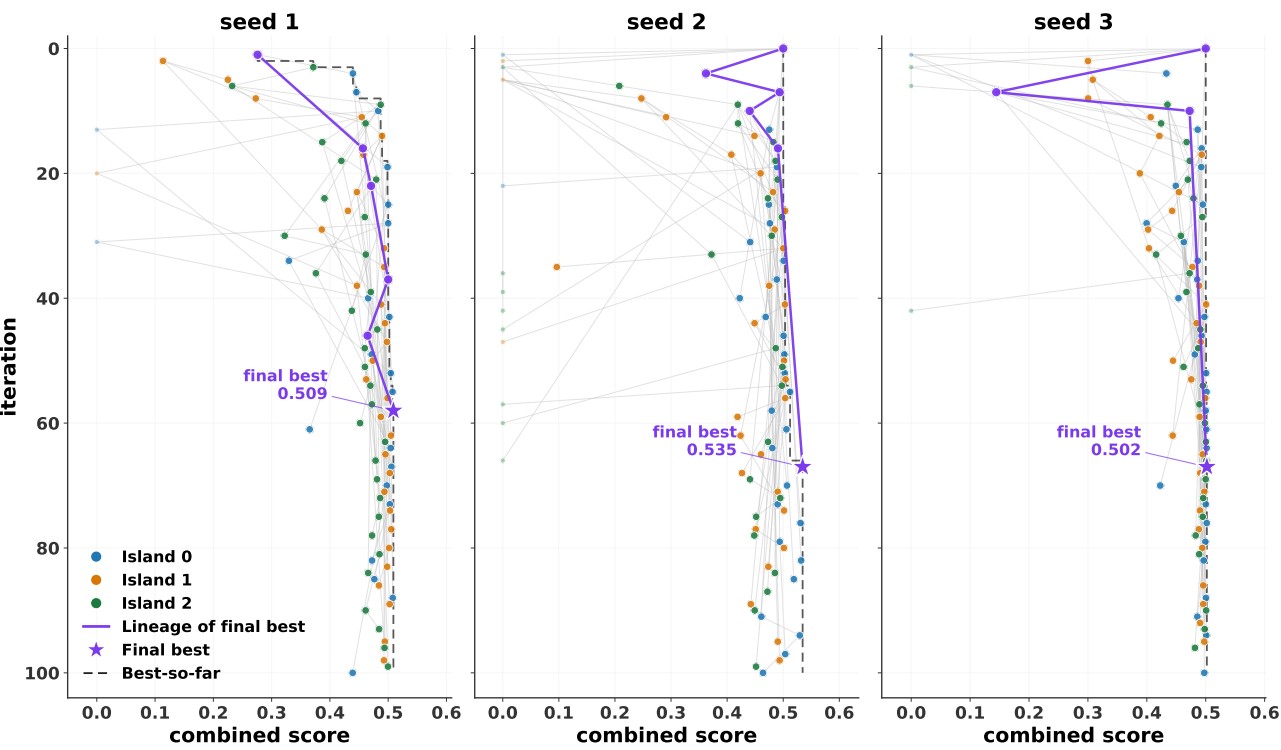

*Figure 19.* Evolution trees for the three `bitbrains_rnd`/H runs. As on `restaurant`, the populations crowd a vertical band around the identity baseline.

## J. Prompt example and reasoning traces

This section demonstrates a working example of EFE-Time on `covid_deaths`. We also provide the EFE-Tab system prompt and it's example final program on `churn` and `in_vehicle_coupon_recommendation`.

### J.1. Time-series example: `covid_deaths`

We pick run 2 of `covid_deaths` and feature the step from iteration 6 to iteration 9 in its lineage (middle panel of Fig. 15, Sec. I; Table 8). The interest of this case is that the search recovers from two consecutive failed mutations (iterations 3 and 6, both with combined score 0) and lands on a robust affine normalization that becomes the structural core of every later program in the lineage. We show the rendered prompt sent to the LLM, the LLM's reasoning, and the resulting program.

*Table 8.* Six-node lineage of the displayed final program for `covid_deaths` run 2. The iteration-9 candidate (the worked example in this section) takes the search from a failure regime back to identity-class performance, after which two further candidates bring the score above the 10%-improvement line.

| Iter. | Combined score | Code lines | Description |
|---|---|---|---|
| 0 | 0.5002 | 85 | identity seed |
| 3 | 0.0000 | 88 | first mutation, high harm rate |
| 6 | 0.0000 | 79 | second failure |
| 9 | 0.4988 | 86 | robust affine normalization (worked example below) |
| 24 | 0.9711 | 108 | branching and winsorisation added |
| 30 | 0.9936 | 108 | final displayed program |

The system prompt establishes the transform-program API, lists transform families, and is held constant across iterations of a run.

The user prompt carries the per-iteration content: a JSON evaluation summary of the parent program, the dataset context for `covid_deaths`, and the evolution history of previously evaluated programs. To keep the section readable, embedded Python code blocks for previously evaluated programs are elided in place; each elision marker reports the line count of the elided body. Section headers, the evaluation-summary JSON, the dataset-context block, and the response template at the end are kept verbatim.

The LLM's response opens with a diagnostic paragraph identifying why the parent failed, states a strategy, and emits the candidate program. We treat the prose preamble as the model's reasoning trace and present the assistant message as a single transcript.

The takeaway from this step is structural: the recent-window affine normalization introduced at iteration 9 is the ancestor of every later program in the lineage. Subsequent iterations refine it (branching for short and constant series, a winsorisation step, a numerical guard) without revisiting the core. For completeness, we list the displayed final program at iteration 30 also below.

## System prompt for EFE-Time

```
You are evolving a target transformation program for time series forecasting.

The program defines a TransformProgram class with three methods:
- fit(y_hist, meta) -> state dict of fitted parameters
- transform(y, state) -> transformed values
- inverse_transform(z, state) -> original-scale values

The downstream forecasting model (Chronos-2) is fixed.
Your job is to find a transform that makes the series easier to forecast.

HOW IT WORKS:
At each forecast origin, the evaluator:
1. Passes the historical target window to fit() along with metadata
2. Transforms the history with transform()
3. Feeds the transformed history to Chronos-2
4. Inverse-transforms the forecast back to original scale
5. Scores MASE against the identity-transform baseline

META DICT AVAILABLE IN fit():
The meta parameter is a dict with these keys:
- prediction_length: forecast horizon (int)
- seasonal_period: e.g. 12 for monthly (int)
- length: total history length
- n_valid: non-NaN count
- mean, std, median: basic location/scale stats
- min_val, max_val, range: value range
- positive_frac: fraction of values > 0
- zero_frac: fraction of values == 0
- skewness: skew of the series
- cv: coefficient of variation (std/|mean|)
- trend_strength: R^2 of linear fit (0 = no trend, 1 = perfect trend)
- recent_mean, recent_std: stats of the last seasonal_period observations

Use these to decide WHICH transform family to use and HOW to parameterize it.
For example:
- High positive_frac + high skewness -> shifted log1p
- Mixed positive/negative + high skewness -> asinh
- Low skewness + low cv -> identity might be best
- Large range + high cv -> scaled transform

TRANSFORM FAMILIES TO EXPLORE:
- identity: no transform (the baseline you must beat)
- shifted log1p: shift = -min + eps, z = log1p(y + shift)
- signed log: z = sign(y) * log1p(|y|)
- asinh: z = asinh(y / scale), with scale fitted from data
- scaled asinh: z = asinh((y - center) / scale)
- bounded power: Box-Cox-like with clamped lambda
- piecewise: different transforms for different value ranges
- affine normalization: (y - center) / scale, e.g. mean/std, median/IQR,
  median/MAD -- robust variants often help when outliers are present
- recent-window normalization: use stats from the last seasonal_period
  observations instead of the full history (useful for non-stationary series)
- detrending: z[i] = y[i] - (slope * i + intercept), fit slope/intercept
  via linear regression. Inverse for the FORECAST must use
  state["hist_len"] + np.arange(len(z)) for the time index, NOT
  np.arange(len(z)) -- see the CRITICAL hist_len section below.
- seasonal phase centering: subtract the mean of each phase i % seasonal_period
  from y. Inverse forecast must use (state["hist_len"] + np.arange(len(z))) % s
  for the phase index, NOT np.arange(len(z)) % s.
- composites: combine any of the above (e.g. detrend -> asinh, or seasonal
  centering -> affine). Each step must be exactly invertible in reverse order.

WHAT TO VARY:
- Family choice based on meta dict
- Shift formula (min-based, percentile-based, etc.)
- Scale formula (std-based, IQR-based, MAD-based)
- Power parameter bounds
- Recent-window vs full-window statistics
- Branching logic (if/else based on meta values)

CRITICAL -- hist_len AND THE FORECAST WORKFLOW:
fit() MUST store "hist_len": len(y_hist) in the returned state dict.
This is critical because transform() and inverse_transform() are called
on DIFFERENT-LENGTH arrays:
- transform() receives the full history (length T)
- inverse_transform() receives the forecast (length prediction_length)
For ANY position-dependent operation (detrending, time-varying scaling),
you MUST use state["hist_len"] to compute the correct time offset for
the forecast positions, NOT len(z). Example for detrending:
  transform:  z[i] = y[i] - slope * i
```

```
    inverse:    y[i] = z[i] + slope * (state["hist_len"] + i)
If you use np.arange(len(z)) in inverse_transform, it will produce
indices 0..11 instead of the correct 72..83, DESTROYING the forecast.

RULES -- YOUR PROGRAM MUST:
- fit parameters using ONLY the history passed to fit()
- ALWAYS include "hist_len": len(y_hist) in the state dict
- NOT use future data
- NOT mutate input arrays (copy first)
- Preserve length, order, and NaN positions
- Return finite values (no Inf); NaN only where input had NaN
- Provide an exact inverse: |inverse(transform(y)) - y| < 1e-4
- Keep inverse stable slightly beyond observed transformed range
- inverse_transform MUST work on arrays of any length (not just hist_len)
- Be deterministic given the same input
- Return a dict from fit() (the state)

COMMON MISTAKES TO AVOID:
- Using np.arange(len(z)) in inverse_transform for position-dependent ops
  (MUST use state["hist_len"] + np.arange(len(z)) for forecast positions)
- Dividing by zero (always add eps to denominators)
- log of negative numbers (check and shift first)
- Inverse that doesn't match forward (verify the math)
- Using np.log instead of np.log1p (log(0) = -inf)
- Forgetting to copy arrays before modifying
- State values that are NaN or Inf

FEEDBACK YOU WILL RECEIVE:
The evaluator provides an "evaluation_summary" artifact with:
- candidate_mase / baseline_mase: your MASE vs identity baseline
- mase_ratio: < 1.0 means you're better than baseline
- help_rate: fraction of series where you're better
Use this to understand WHERE your transform helps and hurts.

STRATEGY:
1. Start by understanding the meta dict values
2. Choose an appropriate transform family
3. Fit parameters carefully from the history using meta dict statistics
4. Verify your inverse math is correct
5. Add branching if a single transform family cannot cover all series
```

## Openevolve user prompt for Covid-Deaths (parent code blocks omitted)

```
# Current Program Information
- Fitness: 0.0000
- Feature coordinates:
- Focus areas: - Fitness unchanged at 0.0000
- No feature coordinates
- Consider simplifying - code length exceeds 500 characters

## Last Execution Output

### evaluation_summary
```
```
{
  "dataset": "covid_deaths",
  "prediction_length": 30,
  "seasonal_period": 1,
  "score": 0.0,
  "baseline_mase": 38.91405167517269,
  "baseline_wql": 0.07616885181861571,
  "baseline_mae": 139.53250150509274,
  "candidate_mase": 51.3413728390549,
  "candidate_wql": 0.16410550044604347,
  "candidate_mae": 337.09089491275,
  "mase_ratio": 1.3193530518902734,
  "wql_ratio": 2.154496182203655,
  "n_valid_series": 223,
  "n_transform_errors": 0,
  "n_inverse_errors": 0,
  "transform_time_sec": 0.16,
  "forecast_time_sec": 0.13,
  "harm_rate": 0.5336322869955157,
  "help_rate": 0.2556053811659193,
  "median_mase_ratio": 1.020853901536389,
  "best_series_ratio": 0.0,
  "worst_series_ratio": 28.91401322388652,
  "complexity": {
    "n_literals": 8,
    "n_branches": 7,
```

```
    "n_fitted_params": 2,
    "penalty": 0.0
  }
}
```

### dataset_context
```
Dataset: covid_deaths
Frequency: D
Prediction length: 30
Seasonal period: 1
Eval series: ?

Sample series statistics (across training samples):
  history length: median=152, range=[152, 152]
  mean: median=23.8, range=[0.987, 4.03e+03]
  std:  median=29, range=[0.987, 4.17e+03]
  min:  median=0, range=[0, 0]
  max:  median=101, range=[2, 9.7e+03]
  positive_frac: median=0.609
  zero_frac: median=0.391
  skewness: median=0.234, range=[-0.040, 1.815]

Description: 266 daily COVID death counts, extreme spikes + regime changes
```

# Program Evolution History
## Previous Attempts

### Attempt 2
- Changes: Full rewrite
- Metrics: combined_score: 0.0000, mase: 49.9797, wql: 0.1596, mae: 285.3394
- Outcome: Mixed results

### Attempt 1
- Changes: Full rewrite
- Metrics: combined_score: 0.0000, mase: 51.3414, wql: 0.1641, mae: 337.0909
- Outcome: Mixed results

## Top Performing Programs

### Program 1 (Score: 0.0000)
```python
# [... 79 lines of program code elided ...]
```
Key features: Performs well on combined_score (0.0000), Performs well on mase (51.3414), Performs well on wql (0.1641), Performs
↪  well on mae (337.0909)

### Program 2 (Score: 0.0000)
```python
# [... 88 lines of program code elided ...]
```
Key features: Performs well on combined_score (0.0000), Performs well on mase (49.9797), Performs well on wql (0.1596), Performs
↪  well on mae (285.3394)

## Inspiration Programs

These programs represent diverse approaches and creative solutions that may inspire new ideas:

### Inspiration 1 (Score: 0.0000, Type: Exploratory)
```python
# [... 88 lines of program code elided ...]
```
Unique approach: Modification:, [Fragment formatting error: 'metric_name'], [Fragment formatting error: 'metric_name']

# Current Program
```python
# [... 79 lines of program code elided ...]
```

# Task
Rewrite the program to improve its FITNESS SCORE.
The system maintains diversity across these dimensions: diversity
Different solutions with similar fitness but different features are valuable.
Provide the complete new program code.

IMPORTANT: Make sure your rewritten program maintains the same inputs and outputs
as the original program, but with improved internal implementation.

```python
```

```
# Your rewritten program here
```

## LLM response for Covid-Deaths (reasoning followed by the candidate program)

Looking at the evaluation results, my current transforms are performing worse than the identity baseline (mase_ratio = 1.32).
↪  The key insight is that this is COVID death data - daily counts with extreme spikes and regime changes. The median skewness
↪  is only 0.234, so most series have moderate skewness.

The problem with my previous approaches:
1. log1p/sqrt transforms compress the recent high values, making it harder for Chronos to extrapolate the current level
2. The scaling in log1p_scaled was distorting the signal
3. The baseline MASE is already 38.9 which is high, meaning this is hard to forecast

Let me try a different strategy: **identity for most series** but with a simple **robust affine normalization** that helps
↪  Chronos by centering/scaling the data to a more "standard" range without distorting the shape. I'll use recent-window
↪  statistics to capture the current regime.

```python
import numpy as np

class TransformProgram:
    def __init__(self, seed=42):
        self.seed = seed

    # EVOLVE-BLOCK-START
    def fit(self, y_hist, meta=None):
        y = y_hist.copy()
        valid = y[~np.isnan(y)]
        state = {"hist_len": len(y_hist)}

        if meta is None:
            meta = {}

        # For COVID deaths: regime changes mean recent stats matter most
        # Use a simple affine normalization based on recent window
        # This preserves shape (no distortion) but helps Chronos with scale

        n = len(valid)
        if n < 5:
            state["family"] = "identity"
            return state

        # Use recent window (last ~30 days) to capture current regime
        recent_n = min(30, n)
        recent = valid[-recent_n:]

        center = float(np.median(recent))
        # Use IQR-based scale for robustness to spikes
        q75 = float(np.percentile(recent, 75))
        q25 = float(np.percentile(recent, 25))
        iqr = q75 - q25

        # MAD as fallback
        mad = float(np.median(np.abs(recent - center)))

        # Pick best scale measure
        if iqr > 1e-6:
            scale = iqr / 1.35  # normalize IQR to std-equivalent
        elif mad > 1e-6:
            scale = mad * 1.4826
        else:
            # Nearly constant recent values - use full history scale or identity
            full_std = meta.get("std", 1.0)
            if full_std > 1e-6:
                scale = float(full_std)
            else:
                state["family"] = "identity"
                return state

        state["family"] = "affine"
        state["center"] = center
        state["scale"] = float(scale)

        return state

    def transform(self, y, state):
        z = y.copy()
        family = state["family"]
```

```
            if family == "affine":
                mask = ~np.isnan(z)
                z[mask] = (z[mask] - state["center"]) / state["scale"]

            return z

        def inverse_transform(self, z, state):
            y = z.copy()
            family = state["family"]

            if family == "affine":
                mask = ~np.isnan(y)
                y[mask] = y[mask] * state["scale"] + state["center"]

            return y
        # EVOLVE-BLOCK-END

def build_transform_program(seed=42):
    return TransformProgram(seed=seed)

def get_transform_program(seed=42):
    return build_transform_program(seed=seed)
```

**Example Final Program for Covid Deaths (reached at iteration 30)**

```python
import numpy as np

class TransformProgram:
    def __init__(self, seed=42):
        self.seed = seed

    # EVOLVE-BLOCK-START
    def fit(self, y_hist, meta=None):
        y = y_hist.copy()
        valid = y[~np.isnan(y)]
        state = {"hist_len": len(y_hist)}
        if meta is None:
            meta = {}
        n = len(valid)
        if n < 10:
            state["family"] = "identity"
            return state

        std_val = meta.get("std", float(np.std(valid)))
        min_val = meta.get("min_val", float(np.nanmin(valid)))
        skewness = meta.get("skewness", 0.0)
        cv = meta.get("cv", 1.0)

        # Detrend using last 45 days - conservative
        trend_n = min(45, n)
        tw = valid[-trend_n:]
        t_idx = np.arange(trend_n, dtype=float)
        t_mean = np.mean(t_idx)
        y_mean = np.mean(tw)
        denom = np.sum((t_idx - t_mean)**2)

        slope = 0.0
        intercept = 0.0
        if denom > 1e-10:
            s = float(np.sum((t_idx - t_mean) * (tw - y_mean)) / denom)
            trend_range = abs(s) * 30
            recent_range = float(np.max(tw) - np.min(tw)) + 1e-6
            # Very conservative: only detrend if extrapolation is safe
            if abs(s) * n > std_val * 0.15 and trend_range < recent_range * 1.0:
```

```python
                    slope = s
                    offset = n - trend_n
                    intercept = float(y_mean - slope * (t_mean + offset))

            state["slope"] = slope
            state["intercept"] = intercept

            v_idx = np.arange(n, dtype=float)
            detrended = valid - (slope * v_idx + intercept)

            recent_n = min(30, len(detrended))
            dt_min = float(np.min(detrended))

            # Use log1p for non-negative skewed data (safer inverse than sqrt)
            if dt_min >= -1e-6 and min_val >= 0 and skewness > 0.6 and cv > 0.5:
                state["nonlinear"] = "log1p"
                log_dt = np.log1p(np.maximum(detrended, 0.0))
                recent_log = log_dt[-recent_n:]
                center = float(np.median(recent_log))
                mad = float(np.median(np.abs(recent_log - center)))
                scale = mad * 1.4826 if mad > 1e-6 else max(float(np.std(recent_log)), 1e-6)
            else:
                state["nonlinear"] = "none"
                recent_dt = detrended[-recent_n:]
                center = float(np.median(recent_dt))
                mad = float(np.median(np.abs(recent_dt - center)))
                scale = mad * 1.4826 if mad > 1e-6 else max(float(np.std(recent_dt)), 1e-6)

            state["family"] = "detrend_affine"
            state["center"] = center
            state["scale"] = max(float(scale), 1e-6)
            return state

    def transform(self, y, state):
        z = y.copy()
        if state["family"] == "identity":
            return z
        mask = ~np.isnan(z)
        idx = np.arange(len(z), dtype=float)
        z[mask] = z[mask] - (state["slope"] * idx[mask] + state["intercept"])
        if state["nonlinear"] == "log1p":
            z[mask] = np.log1p(np.maximum(z[mask], 0.0))
        z[mask] = (z[mask] - state["center"]) / state["scale"]
        return z

    def inverse_transform(self, z, state):
        y = z.copy()
        if state["family"] == "identity":
            return y
        mask = ~np.isnan(y)
        y[mask] = y[mask] * state["scale"] + state["center"]
        if state["nonlinear"] == "log1p":
            # Clamp to prevent exp overflow
            y[mask] = np.clip(y[mask], -10.0, 20.0)
            y[mask] = np.expm1(y[mask])
            y[mask] = np.maximum(y[mask], 0.0)
        idx = state["hist_len"] + np.arange(len(y), dtype=float)
        y[mask] = y[mask] + (state["slope"] * idx[mask] + state["intercept"])
        return y
    # EVOLVE-BLOCK-END

def build_transform_program(seed=42):
    return TransformProgram(seed=seed)
```

```python
def get_transform_program(seed=42):
    return build_transform_program(seed=seed)
```

## J.2. Tabular example: churn

We provide the EFE-Tab system prompt, as well as an example final program learned with EFE-Tab here. The Openevolve prompt is similar to the one for EFE-Time. So we omit this for brevity.

---

**System prompt for EFE-Tab**

```
You are an expert data scientist. You are designing a feature engineering
pipeline for tabular binary classification. Your goal is to improve the AUC
of the model. Think carefully and step-by-step before responding.

You are allowed to create new features from the originals, and to drop the originals.
You are not allowed to train any machine learning model inside fit(),
store a model in the state dict, or mutate the input DataFrame in place.
Do not standardize the data -- it is unnecessary for the models you are using.
Pay attention to the domain context (column names and descriptions), the
decision tree, and the permutation importance reports.
Comment your code to explain your decisions. You should state your reasoning for each decision.

The program you are designing has two methods:
1. fit(X, y) -> state dict -- compute statistics from TRAINING data only
2. transform(X, state) -> DataFrame -- generate features using fitted state

In fit(), compute and store any statistics you need.
In transform(), use the state to build / mutate features.

Allowed operations:
- Arithmetic combining TWO OR MORE columns: sums, differences, products, ratios
- Centered/z-scored interactions using training-fitted statistics from state
- Rank-based interactions using training quantiles from state
- Row-wise aggregates: NaN count, zero count, sum across columns
- Target-rate mapping: per-category target rate from fit() applied in transform()
- Residual features: col_A minus its predicted value from col_B (fitted in fit())
- String/categorical: length, token count, pattern match, category crossing
- Datetime: year, month, day, weekday, elapsed differences
- Parsing string-encoded ordinals into integers  (e.g. "<1" / ">20" -> 0 / 21)
- Dropping an original column you have strong evidence is irrelevant
  (id-like, constant, or flagged as noise/HARMFUL by perm importance)
- Make sure you are not dividing by zero.

FORBIDDEN:
- Training any ML model inside fit() (no LightGBM, no LogReg, no sklearn
  classifiers / regressors, etc.)
- Storing fitted ML models in the state dict (state must remain a plain
  lightweight dict -- see STATE DICT RULES below)
- Modifying an original column in place -- keep originals byte-identical.
  If you want a cleaned/encoded version, add it as a NEW column with a
  different name; drop the original only if you don't need it.
- Mutating the input DataFrame in place -- always copy first

STATE DICT RULES:
- Must be a plain dict (returned from fit())
- Values should be simple types: int, float, str, list, dict, numpy arrays
- Do NOT store DataFrames, models, or non-serializable objects in state

USE THE CONTEXT:
The artifacts include a "dataset_context" with column names, data types,
value ranges, and a dataset description. READ IT CAREFULLY:
- ONLY use column names that appear in the dataset_context
- Read column names to understand what each represents to create
  domain-relevant interactions
- Read the dataset description for domain reasoning, including domain
  quirks
- Check data types before applying operations
```

```
- Use the decision tree to understand which features drive the splits.
  Features near the root are the most important; features that appear
  anywhere in the tree carry signal.
- Use the permutation importance over ORIGINAL features to judge which
  raw columns matter on their own. Consider dropping or replacing any
  flagged as HARMFUL.
- Use the permutation importance over NEW features to judge which of
  your engineered columns earned their cost. Drop new columns flagged as noise or HARMFUL
- Think like an expert data scientist: use the context, domain knowledge, statistics, and
  feedback artifacts to reason about what drives the predictions, then
  engineer accordingly.

FEEDBACK YOU WILL RECEIVE:
1. Dataset context: Column names, dtypes, value ranges, description

2. Stage 2 summary: baseline_cv_auc, candidate_cv_auc, delta_vs_baseline,
   combined_score, AND new_cols / dropped_original_cols lists so you can
   see exactly which original columns were dropped vs kept and which
   fresh columns were added.

3. Decision tree: depth-3 tree on all output features. Features that are closest to the root of the
↪  tree are the most
   important. Features that exist in the tree are also important. Use this to understand which
   features are important and which are not. Use this to understand how to combine features
   to create new features that are important.

4. Permutation importance over ORIGINAL features: AUC drop per raw
   column when shuffled. Computed once at the start of the run.
   USEFUL (>0.005), marginal (0.001-0.005), noise (~0), HARMFUL (<-0.001).
   Drop or replace features labeled HARMFUL. Do not drop features labeled USEFUL, marginal or noise.

5. Permutation importance over NEW features: AUC drop per engineered
   column when shuffled. Computed every evaluation.
   USEFUL (>0.005), marginal (0.001-0.005), noise (~0), HARMFUL (<-0.001).
   Drop NEW features that are noise or HARMFUL.

SCORING:
total_changes = n_drops + n_new_cols
Your score = 0.5 + (CV AUC improvement over baseline) x 5 - 0.10 * sqrt(total_changes / n_rows)
```

## Example final program with EFE-Tab for churn (reached at iteration 33)

```python
"""
Tabular Feature Evolution -- Evolved Feature Program (fit/transform)
"""

import numpy as np
import pandas as pd

class FeatureProgram:
    def __init__(self, seed=42):
        self.seed = seed

    # EVOLVE-BLOCK-START
    def fit(self, X, y):
        state = {}
        global_mean = float(y.mean())
        state['global_mean'] = global_mean

        # Smoothed target encoding for state (51 categories)
        # smooth_factor=30 performed best historically to avoid overfit
        smooth_factor = 30
```

```python
        state_target = {}
        state_col = X['state'].astype(str)
        for cat in state_col.unique():
            mask = state_col == cat
            n = mask.sum()
            cat_mean = float(y[mask].mean()) if n > 0 else global_mean
            smoothed = (n * cat_mean + smooth_factor * global_mean) / (n + smooth_factor)
            state_target[cat] = smoothed
        state['state_target_enc'] = state_target

        return state

    def transform(self, X, state):
        df = X.copy()

        # Drop HARMFUL original: total_eve_charge (perm imp = -0.00179)
        df = df.drop(columns=['total_eve_charge'], errors='ignore')
        # Drop redundant charges (perfectly correlated with minutes)
        df = df.drop(columns=['total_night_charge', 'total_intl_charge'], errors='ignore')

        intl_plan_num = df['international_plan'].astype(str).map({'1': 1, '0': 0}).fillna(0)

        # === PROVEN USEFUL (+0.102): Total charge -- root split in decision tree ===
        # Reconstructs total billing using known per-minute rates
        df['total_charge'] = (df['total_day_charge'] +
                              df['total_eve_minutes'] * 0.085 +
                              df['total_night_minutes'] * 0.045 +
                              df['total_intl_minutes'] * 0.27)

        # === PROVEN USEFUL (+0.025): High service calls binary (tree split at 3.5) ===
        df['high_service_calls'] = (df['number_customer_service_calls'] > 3).astype(int)

        # === PROVEN USEFUL (+0.010): intl_plan x intl_calls ===
        # Low intl calls with intl plan = frustrated user likely to churn
        df['intl_plan_x_intl_calls'] = intl_plan_num * df['total_intl_calls']

        # === From best program (0.5685): intl_plan x service_calls ===
        # Combines the two strongest original predictors
        df['intl_plan_x_svc_calls'] = intl_plan_num * df['number_customer_service_calls']

        # === State target encoding with heavy smoothing ===
        enc = state['state_target_enc']
        gm = state['global_mean']
        df['state_churn_rate'] = df['state'].astype(str).map(enc).fillna(gm)

        # Total: 3 drops + 5 new = 8 changes
        # Penalty: 0.10 * sqrt(8/2333) = 0.0059
        # Removed svc_calls_x_charge (HARMFUL -0.004) and intl_plan_x_day_min (noise)

        return df
    # EVOLVE-BLOCK-END

def build_feature_program(seed=42):
    return FeatureProgram(seed=seed)

def get_feature_program(seed=42):
    return build_feature_program(seed=seed)
```

