# OpenReview forum: "Evolutionary Feature Engineering for Structured Data"
_ICML.cc/2026/Workshop/FMSD — FMSD @ ICML 2026 Poster_

### Official Review · Reviewer_UXV7 · 2026-05-20
**Promising LLM-guided feature engineering framework with useful results across structured data**

**Rating:** 7
**Confidence:** 3

**Review:**

**Summary:** The paper proposes Evolutionary Feature Engineering (EFE), a framework that uses LLM-guided evolutionary search to discover executable preprocessing programs for structured data. The paper instantiates this idea for time-series forecasting through invertible normalization programs, and for tabular prediction through compact feature-engineering programs.

**Strengths:** The paper is interesting and relevant to the workshop. The core idea is simple but useful: instead of changing the downstream model, use LLM-based program search to adapt the input representation. The experimental results are promising, especially the improvements for time-series foundation models and the transfer of learned transformations across forecasters. I also appreciate that the tabular setting includes comparisons to prior LLM-based feature-engineering methods, and that the learned programs are encouraged to remain parsimonious.

**Weaknesses:** I do not see major technical holes. The main limitation is that the evaluation could better isolate where the gains come from. In the time-series setting, the most important comparison is against the identity transform, but stronger comparisons to carefully tuned classical or hand-designed normalizations would make the contribution clearer. The method also appears to depend on a strong frontier LLM and nontrivial search budget, so the practical cost and robustness of the approach should be discussed more explicitly. Finally, the experiments use selected subsets of larger benchmarks, which is understandable for a workshop paper but limits the strength of the empirical claims.

**Suggestions:** Please add more discussion of compute/API cost and sensitivity to the choice of LLM. It would also help to include or discuss stronger non-LLM normalization baselines for EFE-Time. For the tabular experiments, the paper would benefit from a clearer main-text summary, since EFE-Tab is part of the claimed contribution but much of the detail is in the appendix.

**Justification:** Overall, this is a strong workshop submission. The idea is relevant, the implementation is reasonably careful, and the results are promising across both time-series and tabular settings. The limitations are mostly about evaluation scope and practical robustness rather than fundamental flaws.

---

### Official Review · Reviewer_HXLi · 2026-05-20
**LLM-driven evolution of invertible normalizations for TSFMs and parsimonious tabular feature programs**

**Rating:** 7
**Confidence:** 3

**Review:**

## Summary

This paper proposes Evolutionary Feature Engineering (EFE), a framework that uses LLM-driven evolutionary search to discover preprocessing programs for structured data. EFE represents transformations as Python programs with a standardized fit/transform interface and refines them using dataset metadata, summary statistics, and validation feedback. Two instantiations are presented: EFE-Time evolves invertible, dataset-specific normalizations for frozen time-series foundation models, and EFE-Tab evolves compact feature programs for tabular classifiers. On 10 GIFT-Eval datasets, EFE-Time improves Chronos-2 by at least 3% on average across MASE, WQL, and MAE, with gains up to 19% on COVID-Deaths. The learned transforms also transfer to other TSFMs and are additive with fine-tuning. On 9 TabArena datasets, EFE-Tab achieves the best mean rank across feature engineering methods, with especially strong gains for decision trees.

## Strengths

- The idea of evolving invertible normalization programs for frozen TSFMs (EFE-Time) is well-motivated and, to my knowledge, novel. It addresses a real practical gap: fixed normalizations like RevIN or arcsinh cannot adapt to dataset-specific trends, scales, and seasonality.
- Strong experimental design for EFE-Time: transfer across four TSFMs without re-evolution, and the additive gains with fine-tuning (Figure 4) are a convincing demonstration that the learned transforms capture complementary structure
- EFE-Tab's parsimony penalty is a sensible design choice that encourages interpretable programs, and the strong gains for shallow decision trees (Table 2, mean rank 1.39) are practically relevant

## Areas for Improvement

- The reliance on Claude Opus 4.6 raises reproducibility and cost concerns. The local-model ablation (Table 5, Appendix F.8) shows Qwen models are "not stable for non-frontier models," which limits the accessibility of the method. Reporting approximate API cost per dataset would help readers assess practicality
- On 4 of 10 datasets, EFE-Time provides essentially zero improvement (SZ_TAXI, bitbrains_storage, and near-zero on others). The 3% average is driven substantially by COVID-Deaths and Solar. A discussion of when to expect EFE-Time to help versus not would strengthen the paper
- No comparison to fixed normalization baselines (RevIN, arcsinh, z-score) for EFE-Time. The only baseline is the identity transform. Including even one or two standard normalizations would clarify how much of the gain requires evolutionary search versus a well-chosen fixed transform

## Detailed Comments

- The claim "first to evolve invertible time-series normalization modules" is plausible but ELATE (Murray et al., 2025) is cited as closely related. Since no open implementation is available for comparison, this distinction could be stated more carefully
- For EFE-Tab, the comparison against CAAFE and LLM-FE uses only 3 folds from 1 repetition of TabArena due to API costs. This is understandable but limits statistical confidence in the rankings
- The evolution tree visualizations (Appendix I) are a nice touch that helps readers understand the search dynamics

## Justification of Score

This is a well-executed paper with a novel and practically useful contribution in EFE-Time. The invertible normalization search idea is clean, the transfer and additivity experiments are convincing, and the appendix is thorough. The main weaknesses are the reliance on a frontier LLM, the absence of fixed-normalization baselines, and the uneven improvement across datasets. EFE-Tab, while promising, is insufficiently developed in the main text to count as a full contribution. Overall a good workshop paper with clear practical value.

---

### Official Review · Reviewer_v5Nw · 2026-05-22
**Overall, a novel paper with a promising evolutionary approach -- clear accept**

**Rating:** 8
**Confidence:** 4

**Review:**

Summary:
This paper introduces Evolutionary Feature Engineering, a framework that uses LLM-based evolutionary search to discover preprocessing programs for structured data. Programs follow a standardized Python interface and are iteratively refined using dataset context, summary statistics, and validation feedback. The paper focuses on EFE-Time in the main body and defers EFE-Tab and related work to the appendix.

Strengths:
1. Evolving invertible, dataset-specific normalizations for TSFMs is a genuinely novel idea. Prior work on LLM-based time-series feature engineering focuses on generating predictive covariates, not invertible normalizations.
2. The finding that normalizations evolved on Chronos-2 transfer to TimesFM 2.5, Moirai 2, and Reverso-Nano without modification is compelling.
3. The formalization in Section 2 is clear and general. The fit/transform/inverse_transform interface, the scoring function with runtime and complexity penalties, and the evolutionary loop are well-defined and modular. Algorithm 1 is easy to follow.
4. The paper includes full prompt examples, evolution tree visualizations, per-series forecast trajectories showing both successes and failures, and honest analysis of when EFE-Time does not help.

Areas for Improvement:
1. The paper claims three contributions, with EFE-Tab as the third. However, the authors mainly focused on EFE-Tab in the entire appendix. The main text is about EFE-Time only.
2. The authors could provide more comparison against simple normalization baselines for EFE-Time.
3. EFE-Tab is evaluated on 9 binary classification datasets from TabArena. The authors could consider more evaluations on regression tasks, multi-class classification, or larger-scale tabular problems.
4. Could EFE-Time handle channel-dependent or cross-channel normalization?

Detailed Comments:
1. The average improvements of 3.0% MASE, 3.6% WQL, and 3.7% MAE are meaningful but modest. Several datasets show essentially no improvement. The authors could provide more clarification on this.
2. How sensitive is EFE-Time to the number of iterations?

Justification of Score:
The paper contains a genuinely novel core idea but is slightly held back by presentation and evaluation. EFE-Time is novel and compelling, however, the missing standard-normalization baseline is important.